# Annual high-resolution grazing intensity maps on the Qinghai-Tibet Plateau from 1990 to 2020

Jia Zhou[1,2], Jin Niu[3], Ning Wu[1], Tao Lu[1*]

[1]Chengdu Institute of Biology, Chinese Academy of Sciences, Chengdu 610213, China

[2]University of Chinese Academy of Sciences, Beijing 100049, China

[3]Department of Economics, Brown University, Providence, 02912, USA

*Correspondence to*: Tao Lu (lutao@cib.ac.cn)

**Abstract.** Grazing activities constitute the paramount challenge to grassland conservation over the Qinghai-Tibet Plateau (QTP), underscoring the urgency for obtaining detailed extent, patterns, and trends of grazing information to access efficient grassland management and sustainable development. Here, to inform these issues, we provided the first annual Gridded Dataset of Grazing Intensity maps (GDGI) with a resolution of 100 meters from 1990 to 2020 for the QTP. Five most commonly used machine learning algorithms were leveraged to develop livestock spatialization model, which spatially disaggregate the livestock census data at the county level into a detailed 100 m× 100 m grid, based on seven key predictors from terrain, climate, vegetation and socio-economic factors. Among these algorithms, the extreme trees (ET) model performed the best in representing the complex nonlinear relationship between various environmental factors and livestock intensity, with an average absolute error of just 0.081 SU/hm$^2$, a rate outperforming the other models by 21.58%~414.60%. By using the ET model, we further generated the GDGI dataset for the QTP to reveal the spatio-temporal heterogeneity and variation in grazing intensities. The GDGI indicates grazing intensity remained high and largely stable from 1990 to 1997, followed by a sharp decline from 1997 to 2001, and fluctuated thereafter. Encouragingly, comparing with other open-access datasets for grazing distribution on the QTP, the GDGI has the highest accuracy, with the determinant coefficient ($R^2$) exceed 0.8. Given its high resolution, recentness and robustness, we believe that the GDGI dataset can significantly enhance understanding of the substantial threats to grasslands emanating from overgrazing activities. Furthermore, the GDGI product holds considerable potential as a foundational source for other researches, facilitating rational utilization of grasslands, refined environmental impact assessments, and the sustainable development of animal husbandry. The GDGI product developed in this study is available at https://doi.org/10.5281/zenodo.13141090 (Zhou et al., 2024).

## 1 Introduction

Livestock is a crucial contributor to global food systems through the provision of essential animal proteins and fats, and plays a significant role in supporting human survival and socio-economic development (Gilbert et al., 2018; Godfray et al., 2018; Humpenöder et al., 2022; Kumar et al., 2022). However, the escalating increase in human demand for meat and dairy products over recent decades has triggered a livestock boom, which in turn has increasingly threatened grassland ecosystems and placed a heavy burden on the environment through overgrazing and land-use change (Tabassum et al., 2016; Wei et al., 2022; Minoofar et al., 2023). It is estimated that up to 300 million hectares of land are used globally for grazing and cultivating fodder crops (Tabassum et al., 2016). Grazing activities could alter vegetation phenology and community structure (Dong et al., 2020), and trigger deforestation (García Ruiz et al., 2020), grassland degradation (Sun et al., 2020), soil erosion (Shakoor et al., 2021), and associated direct releases in greenhouse gas that lead to climate change feedback (Godfray et al., 2018; Chang et al., 2021). Additionally, livestock are responsible for large-scale dispersion of pathogens, organic matter, and residual medications into soil and groundwater, thereby contaminating the environment (Venglovsky et al., 2009; Tabassum et al., 2016; Hu et al., 2017; Muloi et al., 2022). Consequently, more and more scholars have called attention to provide reliable contemporary dataset to illustrate the spatio-temporal heterogeneity and variation of livestock (Petz et al., 2014; Fetzel et al., 2017; Zhang et al., 2018; Li et al., 2021).

One of the major challenges in monitoring grazing activity at regional or even larger scale, is the determination of the livestock distribution pattern. Despite the importance of geographical grazing information, high spatio-temproal grazing dataset remain unavailable, posing the most critical challenge to grassland management, particularly for vulnerable grassland ecosystems in fragile regions grappling with economic and sustainable development contradictions (Meng et al., 2023; Pozo et al., 2021; Miao et al., 2020; He et al., 2022). In the early 2000s, the Food and Agriculture Organization of the United Nations (FAO) launched the Gridded Livestock of the World (GLW) project to facilitate a detailed evaluation of livestock production, aiming to provide pixel-scale livestock densities instead of traditional administrative unit benchmarks (Nicolas et al., 2016). Consequently, the world's inaugural dataset of livestock spatialization map (GLW1) was released in 2007, providing the first globally standardized livestock density distribution map at a spatial resolution of 0.05 decimal degrees (≈5 km at the equator) for 2002. It was not until 2014 that an updated GLW2 map with a 1 km resolution for 2006 was released, by using a stratified regression approach, superior spatial resolution predictor variables, and more detailed livestock census data (Robinson et al., 2014). Furthermore, an evolutionary step in machine learning technology saw Gilbert et al. (2018) using random forests algorithm to forge a global livestock distribution map with a 10-km resolution for 2010 (GLW3), succeeding traditional multivariate regression methods and surpassing the precision of previous GLW1 and GLW2 maps. Beyond these global mappings, several maps with different scales have also been published, including intercontinental, national, state or provincial, and local scale (Neumann et al., 2009; Prosser et al., 2011; Van Boeckel et al., 2011; Nicolas et al., 2016). However, these maps are fundamentally coarse due to constraints such as the availability of fine scale and contemporary census data, the grazing spatialization method, as well as the identification of appropriate indicators, thereby limiting their application to local or regional-scale studies (Nicolas et al., 2016; Gilbert et al., 2018; Robinson et al., 2014). Hence, there is an emergent demand for more refined grazing map products (Mulligan et al., 2020; Martinuzzi et al., 2021).

An exemplar of this need can be observed in the Qinghai-Tibet Plateau (QTP), the world's most

elevated pastoral region and an important grazing area in China (Zhan et al., 2023). It was possessing abundant grassland that spans 1.5 million km$^2$, accounting for 50.43% of China's total grassland area, with Yak and Tibetan sheep as primary grazing livestock (Feng et al., 2009; Cai et al., 2014; Zhan et al., 2023). Over recent decades, the QTP has undergone escalating grassland degradation, leading to many ecological and socio-economic problems, which calls for an urgent need for detailed livestock distribution dataset (Li et al., 2022a). Unfortunately, despite researchers' efforts at mapping the QTP's grazing intensity, current livestock dataset still suffer from coarse spatio-temporal resolution and modelling accuracy. Apart from the aforementioned global grazing dataset, several other maps also cover the QTP. For instance, Liu et al. (2021) generated annual 250-m gridded carrying capacity maps for 2000-2019, by employing multiple linear regressions of livestock numbers, population density, NPP, and topographic features. Li et al. (2021) used machine learning algorithms to produce gridded livestock distribution data at 1 km resolution for 2000-2015 in western China at five year interval, based on county-level livestock census data and 13 factors from land use practice, topography, climate, and socioeconomic aspects, including grassland coverage, arable land coverage, forest land coverage, desert coverage, NDVI, elevation, slope, daytime surface temperature, precipitation, distance to river, travel time to major cities, population density, and GDP (Li et al., 2021). A contribution from Meng et al. (2023) brought forth annual longer time-series grazing maps by using random forests model, integrating climate, soil, NDVI, water distance, and settlement density to decompose county-level livestock census data to a 0.083° (≈10 km at the equator) grid for 1982-2015 (Meng et al., 2023). Similarly, Zhan et al. (2023) also used random forests algorithm to combine eleven influence factors to provide a winter and summer grazing density map at 500 m resolution for 2020 (Zhan et al., 2023).

However, although these maps have provided good help in understanding grazing conditions on the QTP, there are currently still no maps that can satisfy the need for fine-scale grassland management with a long time span. In addition, the available livestock distribution maps of the QTP still need improvement in terms of modelling techniques and factor selection to obtain high-precision livestock spatialization data. For example, traditional methods like multiple linear regression, while proven fundamental and widely applicable for livestock spatialization (Robinson et al., 2014; Ma et al., 2022), are being challenged by the development of computational science in recent years. Among them, machine learning technology is providing new opportunities towards more accurate predictions of livestock distribution (García et al., 2020). Random forests regression, for instance, is currently widely used to construct global, national as well as regional livestock spatialization dataset, and has been proved to have much better accuracy than traditional mapping techniques (Rokach, 2016; Nicolas et al., 2016; Gilbert et al., 2018; Dara et al., 2020; Chen et al., 2019; Li et al., 2021). Nevertheless, other more advanced machine learning methods with superior feature learning and more robust generalization capabilities, remains largely untapped for modelling geographic data (Ahmad et al., 2018; Heddam et al., 2020; Long et al., 2022). Thus, exploring the potential application of new advanced machine learning technologies in livestock spatialization remains a critical task. Furthermore, selecting the suitable factors that influencing livestock grazing preferences is also the other critical challenge for enhancing the precision of grazing distribution dataset (Meng et al., 2023). Livestock grazing activities are often affected by abiotic and biotic resources, including climatic and environmental factors (Waha et al., 2018), herd foraging and grazing behaviours (Garrett et al., 2018; Miao et al., 2020), and conservation-oriented policies (Li et al., 2021). For instance, regions exceeding elevations of 5,600 m or slope greater than 40% are customarily unsuitable for grazing (Luo et al., 2013; Mack et al., 2013; Robinson et al., 2014; Chen et al., 2019). Moreover, the livestock generally prefer areas abundant in

water and pasture resources for foraging (Li et al., 2021). Besides, ecological conservation policies also exert substantial influence, significantly affecting grazing distribution relative to the level of conservation priority. In addition, the health status of the grassland is an important factor influencing whether livestock choose to feed or not (Li et al., 2021). Consequently, indicators related to the above aspects are often employed to gauge the spatial heterogeneity of livestock distribution (Allred et al., 2013; Sun et al., 2021; Meng et al., 2023). Nonetheless, some most commonly used indicators like NPP or NDVI can result in misconceptions, as they may not fully characterize the grazing intensity. For example, grasslands with high NPP or NDVI are often preferred by livestock, but this doesn't necessarily correlate with grazing intensity in nature reserves due to strict policy restrictions (Veldhuis et al., 2019; O'neill and Abson, 2009; Zhang et al., 2021b). Conversely, areas with sparse grassland cover may support considerable livestock numbers, despite evidence of degradation (Zhang et al., 2021a; Guo et al., 2015). Accordingly, further investigation of novel indicators is imperative to enhance the correlation between grassland and grazing intensity, thereby optimizing the integration of such influencing factors into grazing spatialization models.

In summary, the QTP is in pressing need for a high spatio-temporal resolution grazing dataset to address urgent and realistic challenges. But the existing livestock dataset specific to the QTP are fraught with several insufficient, predominantly concerning rough resolution, relatively backward census data, as well as conventional methods in livestock spatialization. Moreover, the discrepancies in predictive indicators and modelling approaches within these dataset discourage their application in time-series analysis. Consequently, the generation of high-resolution and high-quality grazing map products has emerged as the most pressing challenge for the QTP. Here, we aim to (1) establish a methodological framework by using more rational models and indicators than traditional studies to achieve fine-scale livestock spatialization; (2) select the grazing spatialization model with good performance by incorporating multi-source data with advanced machine learning techniques; and (3) ultimately, provide an annual grazing intensity dataset with 100 m resolution spanning from 1990-2020. These maps can not only provide fundamental dataset with finer spatio-temporal resolution to address the limitations of existing grazing intensity maps, but enhance a better understanding of sustainable management practices as well as other grassland-related issues across the QTP.

**2 Data and methods**

**2.1 Study area**

Known as the Asia's water tower and the world's third pole, the QTP is geographically situated between 73°19~104°47′ east longitude and 26°00′~39°47′ north latitude, with a total area of about 2.61 million square kilometers (Figure 1). Its jurisdiction encompasses 182 counties within six provincial regions of China, including Tibet Autonomous Region, Qinghai Province, Xinjiang Uygur Autonomous Region, Gansu Province, Sichuan Province, and Yunnan Province (Meng et al., 2023). Elevation on the QTP predominantly ranges between 3,000 m and 5,000 m, with an average altitude exceeding 4,000 m. With grasslands constituting over half of its land cover, the QTP emerges as one of the most important pastoral areas in China. Alpine steppe, alpine meadow, and temperate steppe characterize the main grassland types on the QTP (Han et al., 2019; Zhai et al., 2022; Zhu et al., 2023b). The complex geographical and climatic conditions of the QTP contributes to the markedly heterogeneous grassland distribution, which correspondingly lead to the high heterogeneity in livestock distribution. Moreover, social and economic development, coupled with policy initiatives directed towards grassland restoration,

have noticeably impacted the livestock numbers on the QTP over recent decades (Li et al., 2021; Li et al.,
2016).

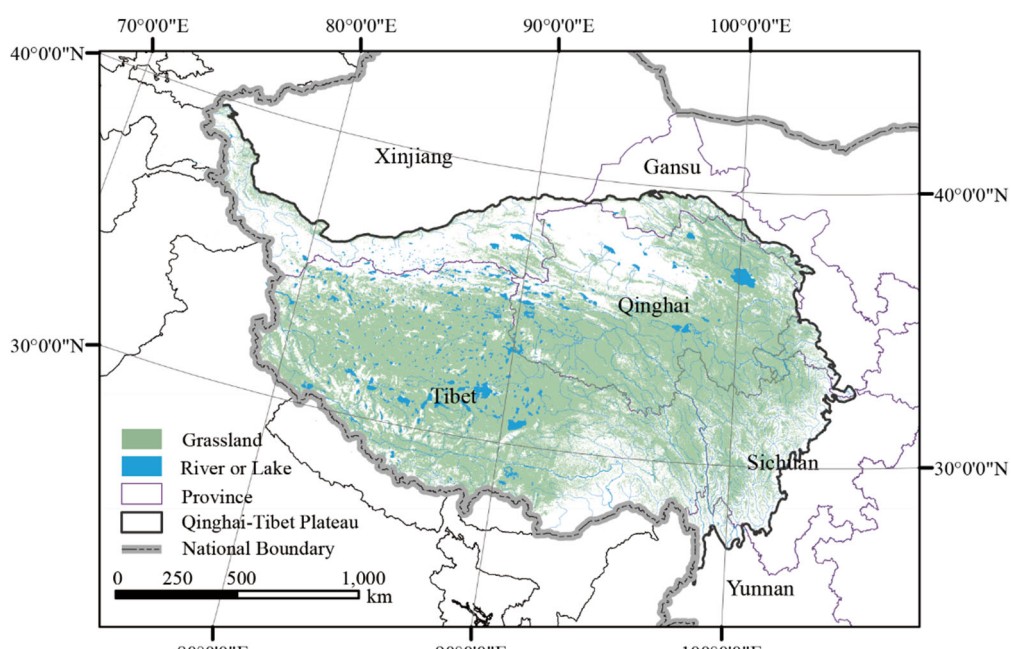

Figure 1. The geographic zoning map of the Qinghai-Tibet Plateau (QTP) superposed with grassland vegetation.
Boundaries for the six provinces used for statistical analysis are also shown.

### 2.2 Data source

*2.2.1 Census livestock data*

The county-level census livestock data for the period between 1990 and 2020 were obtained from
the Bureau of Statistics of each county across the QTP (Table 1). The data includes the number of cattle,
sheep, horse and mule, with the exception of counties in Yunnan Province, which lack data for the
168  years from 1990 to 2007, and Ganzi Prefecture in Sichuan Province, which lack data for the years from
1990 to 1999, and Muli county in Sichuan Province, which lack data for the years from 1990 to 2007.
For these counties belonging to the same prefecture, including counties in Ganzi and Aba prefectures in
Sichuan Province, we used the livestock census data at the prefecture-level to carry out spatialization.
For these counties in Yunnan Province, since they belong to different municipalities, it is not reasonable
to replace them with municipal-level data. For these counties without livestock census data for some
174  years, we supplemented the missing data by linear interpolation with grazing density data in available
175  year. In total, livestock data were available for 182 counties, and 4,998 independent records were
finally generated. Furthermore, the respective quantities of different livestock types are converted to
Standard Sheep Units (SU), in compliance with the Chinese national regulations (Meng et al., 2023).

Due to the difficulty of collecting township-level census livestock data, the validation data at the
township scale collected in this study only involved these townships of Baching County (2010-2018)
and Gaize County (2018-2020) in Tibet, and Hongyuan County in Sichuan Province (2008). The
township-level census livestock data cumulatively involves 18 townships with a total of 112 records,
and were only used for auxiliary validation of the simulation results.

The validation data at the pixel scale also encompass a total of 112 records from 68 sites, which
were collected from literatures, questionnaires and field surveys. Specifically, 93 records at 49 sites

spanning the 1990-2021 period were obtained from 17 literatures, 19 records at 19 sites were obtained from the questionnaires and the field survey in 2021. The detailed information for these records can be found in the Supplementary files (Figure S3 and Table S3).

Table 1. Summary of the livestock data used in this study

| Variables | Scale | Time | Sources |
|---|---|---|---|
| Livestock numbers | County | 1990-2020 | Statistical bureau |
| | Township | 2008-2020 | Statistical bureau |
| | Pixel | 1990-2021 | Literatures, questionnaires and field surveys |

*2.2.2 Factors affecting grazing activities*

Livestock grazing activities are often affected by abiotic and biotic resources, including climatic and environmental factors (Waha et al., 2018), herd foraging and grazing behaviours (Garrett et al., 2018; Miao et al., 2020). For instance, high-altitude and steep hillsides are unsuitable for grazing due to terrain constraints, and the distribution of herders directly affects the grazing areas (Luo et al., 2013; Mack et al., 2013; Robinson et al., 2014; Chen et al., 2019). Moreover, the livestock generally prefer areas abundant in water and pasture resources for foraging (Li et al., 2021). Therefore, in this study, topography, climatic, environmental and socio-economic impacts were considered as influential factors on grazing activities (Li et al., 2021; Meng et al., 2023).

Table 2. Summary of factors affecting grazing activities on the QTP.

| Variables | Format | Period | Time Resolution | Spatial Resolution | Source |
|---|---|---|---|---|---|
| Altitude | GeoTIFF | —— | —— | 30m | https://www.gscloud.cn |
| Slope | GeoTIFF | —— | —— | 30m | https://data.tpdc.ac.cn |
| Water source | Shapefile | 1990-2020 | Annual | —— | https://data.tpdc.ac.cn |
| Population density | GeoTIFF | 1990-2020 | Annual | 100m | See supplementary file |
| Temperature | GeoTIFF | 1990-2020 | Annual | 100m | See supplementary file |
| Precipitation | GeoTIFF | 1990-2020 | Annual | 100m | See supplementary file |
| HNPP | GeoTIFF | 1990-2020 | Annual | 100m | See supplementary file |

We utilized correlation analysis and the Random Forest importance ranking tool to eliminate redundant environmental factors and determine the contribution of each factor. Ultimately, altitude, slope, distance to water source, population density, air temperature, precipitation and human-induced impacts on NPP (HNPP) was selected as indicators (Table 2). Specifically, elevation is derived from the DEM dataset accessible via the Resource and Environmental Data Cloud Platform of the Chinese Academy of Sciences (https://www.gscloud.cn), which also facilitated slope calculation. Rivers and lakes were obtained from the National Tibetan Plateau Data Center (https://data.tpdc.ac.cn), and the nearest Euclidean distance from each pixel to rivers or lakes is calculated accordingly. Meteorological elements such as daily air temperature and precipitation were downloaded from the China Meteorological Data Service Center (http://data.cma.cn). For the grid dataset that is not conditionally available, including population density, temperature, precipitation and HNPP, we detailed the creation process in the Supplementary file. All datasets utilized in this study were harmonized to consistent coordinate systems and resolutions (WGS 1984 Albers, 100 m).

## 2.3 Methodological framework

We adopted a comprehensive methodological framework for mapping high-resolution grazing intensity on the QTP. Three major steps are included to predict the distribution pattern of grazing intensity: (1) identifying factors affecting grazing activities and extracting theoretical suitable areas for livestock grazing, (2) building grazing spatialization model, and (3) filtering the model and correcting the grazing map. An exhaustive explanation of each step is provided in Figure 2.

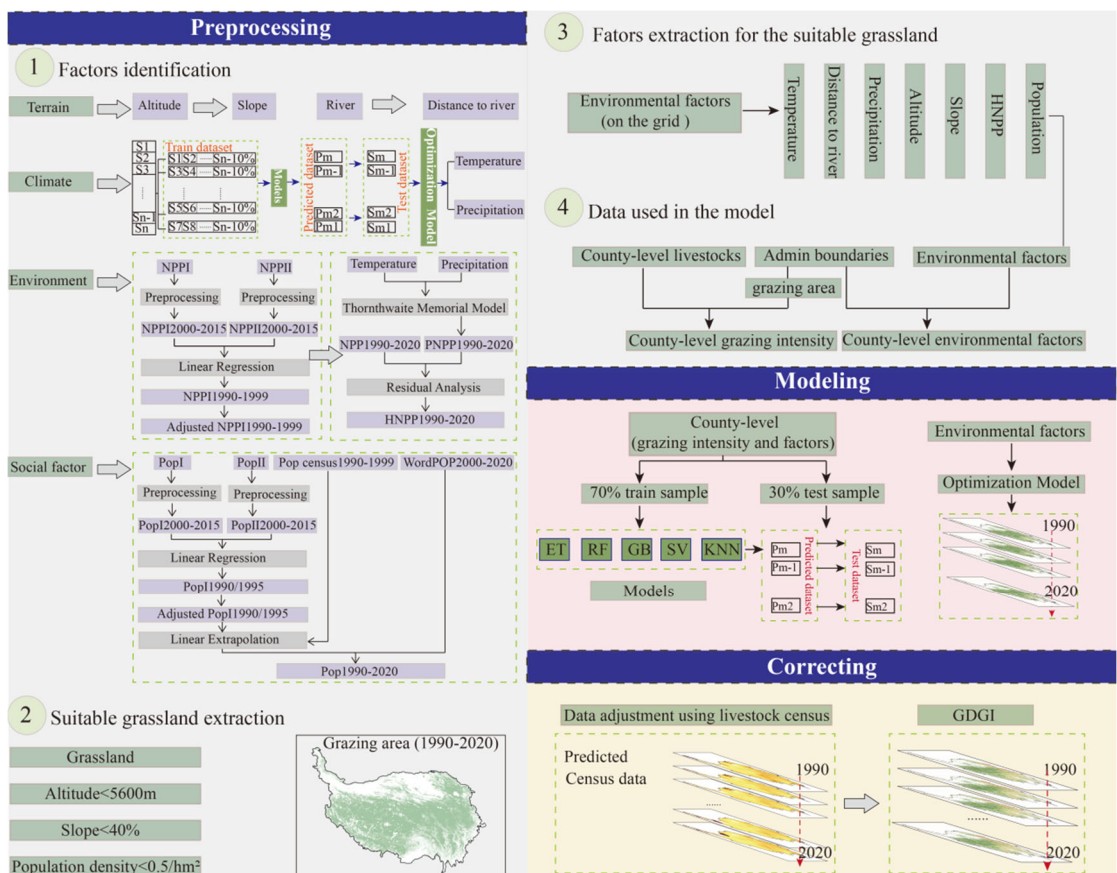

Figure 2. Flowchart of creating grazing intensity maps using different methods and source products.

### 2.3.1 Identifying factors and theoretical suitable areas for grazing

In this study, we assumed that grazing activities are confined solely to grassland. Consequently, the potential grazing areas for each year were identified on the basis of grassland boundaries, which was extracted from the 30 m annual land cover dataset (CLCD) (Yang and Huang, 2021). Furthermore, grassland with slope over 40% and elevation higher than 5,600 m respectively, were considered unsuitable for grazing and were therefore excluded from the potential grazing area in the subsequent simulations (Robinson et al., 2014). In addition, the grassland with population density greater than 50 inhabitants km$^{-2}$ were also excluded (Li et al., 2018). The remaining isolated grassland was thus categorized as theoretical feasible grazing regions.

The spatial patterns of abiotic and biotic resources, incorporating food availability, environmental stress, and herder preference critically affect grazing activities (Meng et al., 2023). In light of this, seven influencing factors in four aspects were selected for grazing intensity mapping (Figure 2-1).

 *2.3.2 Building grazing spatialization model*

By performing regional statistics, the annual average values for each grazing influence factor were extracted from the theoretically suitable grazing areas at the county scale, and were further used as independent variables in the model construction. The dependent variable for the model was acquired by determining the livestock density within each county, followed by a logarithmic transformation of the values to normalize the distribution of the dependent variable. Consequently, a total of 4,998 samples were derived from the aforementioned independent and dependent variables. Of these samples, 70% were allocated for model training, while the remaining 30% comprised the test sets, serving to validate the model's performance. Subsequently, we built grazing spatialization models using five machine learning algorithms at the county scale, including Support Vector regression (SV) (Cortes and Vapnik, 1995; Lin et al., 2022), K-Nearest Neighbors (KNN) (Cover and Hart, 1967), Gradient Boosting regression (GB) (Friedman, 2001; Pan et al., 2019), Random Forests (RF) (Breiman, 2001) and Extra Trees regression (ET) (Geurts et al., 2006; Ahmad et al., 2018) (see Supplementary file for details). Lastly, to assess the accuracy of the spatialized livestock map, the predicted livestock intensity values were juxtaposed with the livestock statistical data from each respective county.

*2.3.3 Correcting the grazing map*

We further used the optimal model to predict the geographical distribution of grazing density across the QTP. To maintain better consistency between the predicted livestock number and the census data, the estimated results were adjusted using the census livestock numbers at the county scale as a control according to Equation (1). Consequently, the corrected and refined map is presented as the final grazing intensity map in this study.

$$L_{correction} = \frac{L_{CCensus}}{L_{Cgrid}} \times L_{grid} \qquad (1)$$

where $L_{correction}$ is the predicted pixel-scale livestock number after adjustment, $L_{Cgrid}$ represents the estimated livestock number for each county, $L_{CCensus}$ is the census livestock number for each county, and $L_{grid}$ refers to the predicted livestock number at the pixel scale.

**2.4 Accuracy evaluation**

We used three accuracy validation indexes to evaluate the performance of five machine learning algorithms, including coefficients of determination ($R^2$), mean absolute error (MAE), and root mean square error (RMSE), by through a comparison of the predicted value with the census data. The definitions of three metrics are presented in Equation (2) to (4).

$$R^2 = 1 - \frac{\sum_{i=1}^{n}(C_i - P_i)^2}{\sum_{i=1}^{n}(C_i - \overline{C})^2} \qquad (2)$$

$$MAE = \frac{1}{n}\sum_{i=1}^{n}|C_i - P_i| \qquad (3)$$

$$RMSE = \sqrt{\frac{1}{n}\sum_{i=1}^{n}(C_i - P_i)^2} \qquad (4)$$

where $C_i$ and $P_i$ are the census livestock data and the predicted value for county $i$, respectively; $\overline{C}$ represents the mean census value for all county; and $n$ gives the total number of counties.

**2.5 uncertainties evaluation**

Uncertainty in our grazing intensity maps can stem from multiple sources, such as the constraints of
cross-scale modeling and the intrinsic inaccuracies of the input data. To quantify these uncertainties, we
utilized the Monte Carlo (MC) method, conducting 100 iterations of simulation. Subsequently, we
evaluated uncertainty through the Mean Relative Error (MRE) and assessed the model's robustness
using the Standard Deviation (STD), following established methodologies (Yang et al., 2020;
Alexander et al., 2017; Mcmillan et al., 2018). The definitions for these metrics are delineated in
Equations (5) to (7).

$$\text{MC} = \frac{1}{n}\sum_{i=1}^{n} f(x_i) \tag{5}$$

$$\text{MRE} = \frac{1}{n}\sum_{i=1}^{n}\left|\frac{x_i - \bar{x}}{\bar{x}}\right| \tag{6}$$

$$\text{STD} = \frac{1}{n}\sum_{i=1}^{n} f(x_i)\sqrt{\frac{1}{n}\sum_{i}^{n}(x_i - \bar{x})} \tag{7}$$

where $x_i$ are random samples, $f(x_i)$ is the function evaluated at $x_i$, and $n$ is the number of
simulations. $\bar{x}$ represents the mean value for all simulation maps.

**3 Results**

**3.1 Performances of models**

Table 3 summarizes the efficiency of the five used machine learning models with considering all
three accuracy evaluators of $R^2$, MAE and RMSE. It can be seen that the ET model performs the best,
with its $R^2$ exceeding 0.955, and MAE (0.081 SU/hm$^2$) and RMSE (0.164 SU/hm$^2$) significantly lower
than the value of RF, GB, KNN and SVM models. Figure 3 illustrates the correlation between the
census livestock data and the livestock numbers predicted by the model for each county from 1990 to
2020. It demonstrated that the ET-predicted data displayed a distribution pattern consistent with that of
other models, but the scatter points of the ET model were more convergent to the 1:1 diagonal line,
indicating a superior fit compared to the other models. These comparisons suggest that the ET model
possesses superior robustness and can, therefore, provide stable estimations of livestock intensity on
the QTP.

Table 3. Comparison of mapping accuracy for five machine learning models based on the same validation datasets

| Models | $R^2$ | MAE (SU/hm$^2$) | RMSE (SU/hm$^2$) |
|--------|-------|-----------------|------------------|
| ET | 0.955 | 0.081 | 0.164 |
| RF | 0.928 | 0.099 | 0.208 |
| GB | 0.859 | 0.197 | 0.300 |
| KNN | 0.786 | 0.186 | 0.384 |
| SVM | 0.380 | 0.419 | 0.750 |

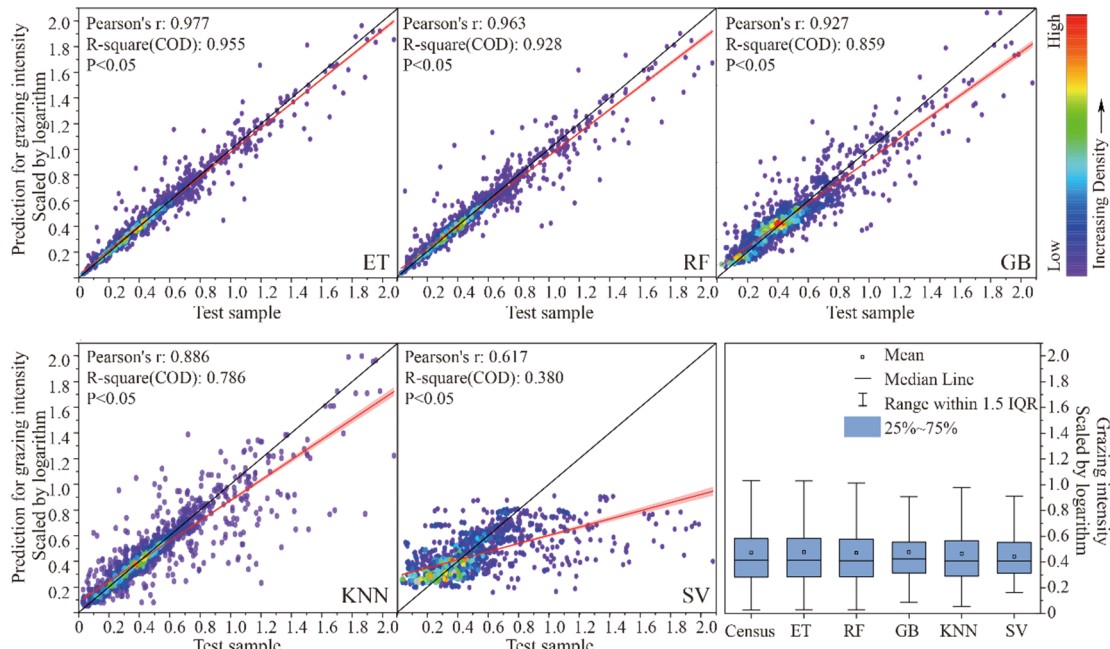

Figure 3. Scatterplots of model-predicted livestock numbers and census grazing data at the county scale. The red

solid line and the black solid line are the fitting line and the 1:1 diagonal line, respectively.

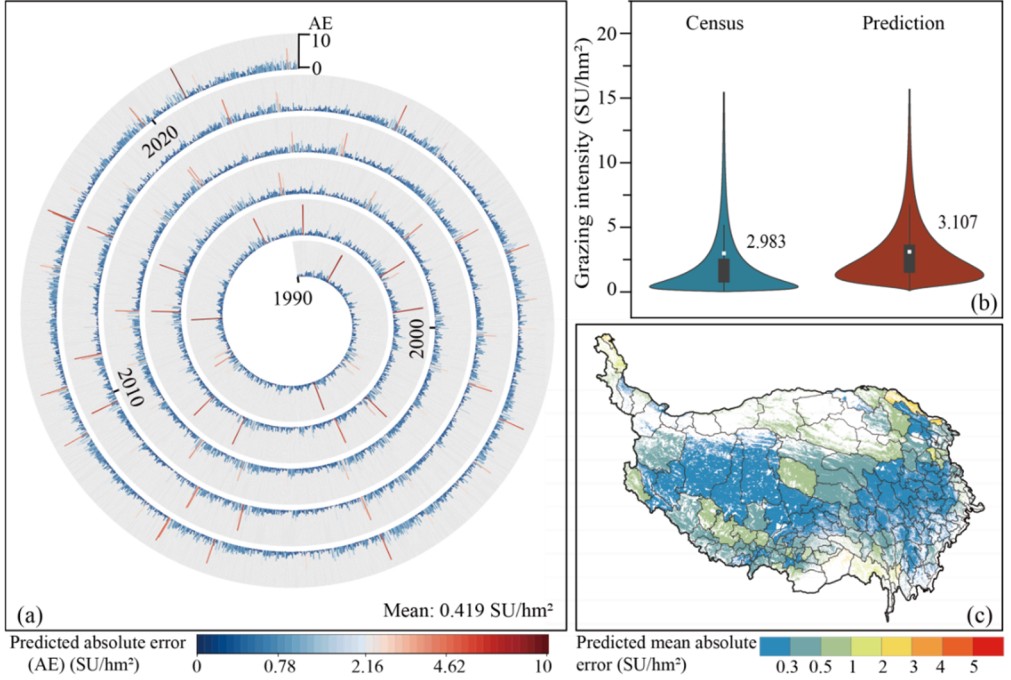

Figure 4. Accuracy of the ET-predicted grazing intensity results at spatial resolution of 100 m from 1990 to 2020.

(a) absolute error (AE) between the predicted and the census data at the county scale from 1990 to 2020; (b)

comparison of the predicted and census data of the whole QTP from 1990 to 2020; (c) spatial distribution of the

mean absolute error (MAE) during 1990 to 2020 for each county.

Using the ET model, we projected the spatio-temporal distribution of grazing intensity across the

QTP from 1990 to 2020 at a 100 m × 100 m resolution. To validate the accuracy of these predictive

maps, we upscaled the pixel-level predictions to the county level and compared them against livestock

census data (Figures 4a and 4b). The results clearly show a high degree of consistency between the

predicted livestock intensity and the county-level census data, especially in areas with lower grazing

intensity (Figures 4a and 4b). Specifically, while the mean census data indicated 2.983 SU/hm² for

livestock intensity, our county-level predictions yielded an average of 3.106 SU/hm², with a MAE of

0.123 SU/hm², a RMSE of 0.580 SU/hm², and an R² value of 0.669. Additionally, 76.31% of the

counties (n=3,814) exhibited data discrepancies of no more than 0.6 SU/hm², and 91.74% (n=4,585)

had discrepancies under 1.0 SU/hm². Regarding spatial distribution, areas with data discrepancies of

less than 0.3 SU/hm² were predominantly located in the northwest and southeast regions of the QTP. In

certain counties of the northeast and southwest, the variations were even below 1.0 SU/hm² (Figure 4c).

## 3.2 Evaluation of uncertainties

We have chosen the Mean Relative Error (MRE) as a key metric for evaluating the simulation

accuracy of grazing intensity within the QTP. Employing Monte Carlo simulations spanning the period

from 1990 to 2020, our research findings demonstrate that the average MRE for grazing intensity

across the QTP ranged between 6.84% and 9.08% (Figure 5a). The spatial distribution of MRE

indicates that the majority of the plateau exhibits low error margins. For example, in 2020, areas with

an MRE of less than 5% accounted for 35.86% of the total grassland area, while those with an MRE

below 10% constituted 75.84%. Only 3.38% of the grasslands had an MRE exceeding 20%, with these

regions primarily located in the southwestern portion of the QTP (Figure 5b). Moreover, the robustness

analysis suggests that the majority of regions within the QTP display relatively stable grazing intensity

trends. For instance, the overall standard deviation (STD) in 2020 was 0.059 SU/hm², with the

northwest region demonstrating remarkable stability, reflected in an STD of less than 0.005 SU/hm².

Although some areas within the Yarlung Zangbo River Basin and the eastern part of Qinghai Province

experienced higher variability, their STD was still maintained below 0.3 SU/hm² (Figure 5c).

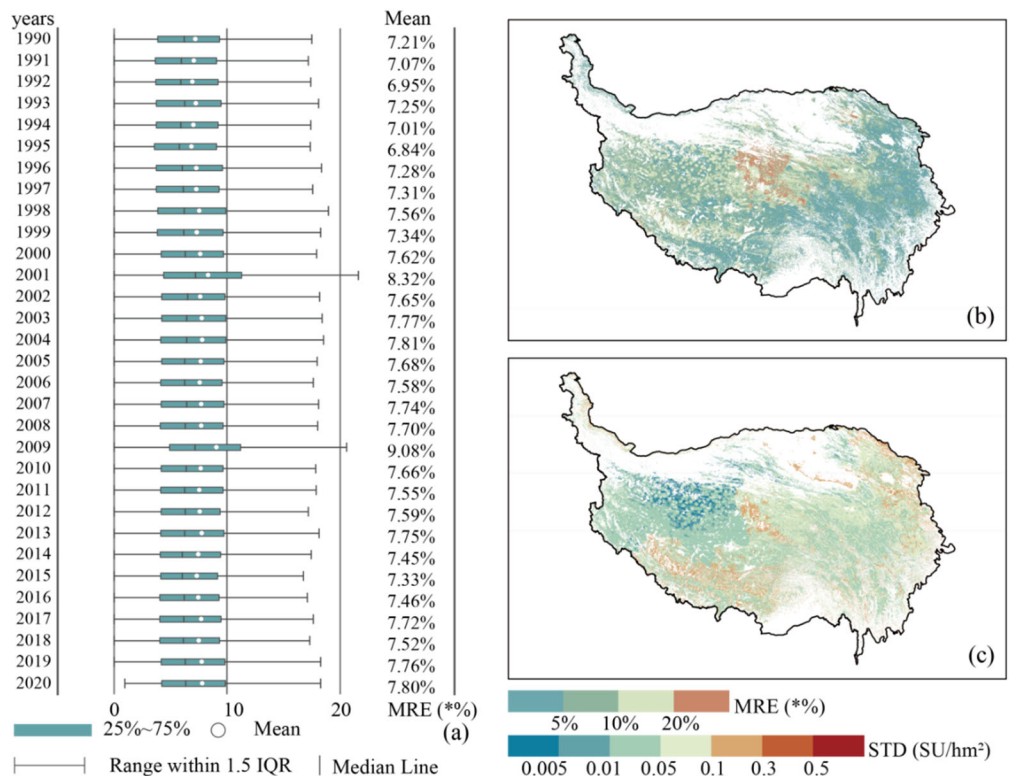

Figure 5. Uncertainty analysis of grazing intensity maps based on ET and Monte Carlo methods. (a) MRE of

grazing intensity maps from 1990 to 2020, (b) spatial distribution of MRE, (c) spatial distribution of STD.

## 3.3 Validation of the GDGI dataset

After employing county-level livestock census as a benchmark for quality control, we obtained the annual Gridded Dataset of Grazing Intensity maps (GDGI) across the QTP spanning 31 years from 1990 to 2020. We firstly confirmed the accuracy of the GDGI dataset based on 112 field grazing intensity records at 68 sites (see Table S3 in Supplementary file for details), which ranged from 0 to 5.61 sheep unit per hectare (SU/hm$^2$), and covered three main grasslands on the QTP: the alpine steppe (N=62), alpine meadow (N=46), and alpine desert steppe (N=4). The GDGI dataset was assessed by undertaking a comparative accuracy assessment between it and the field grazing intensity data (Figure 6a). It can be seen that in general, our dataset was highly consistent with the reference ground-truth validation data, with $R^2$ = 0.804, MAE = 0.572 SU/hm$^2$, and RMSE = 0.953 SU/hm$^2$. Moreover, the absolute errors between the GDGI data and the field grazing intensity data were relatively small, with more than half of the records having an error below 0.3 SU/hm$^2$, 78.57% below 1.0 SU/hm$^2$, and 89.29% below 1.5 SU/hm$^2$ (Figure 6b).

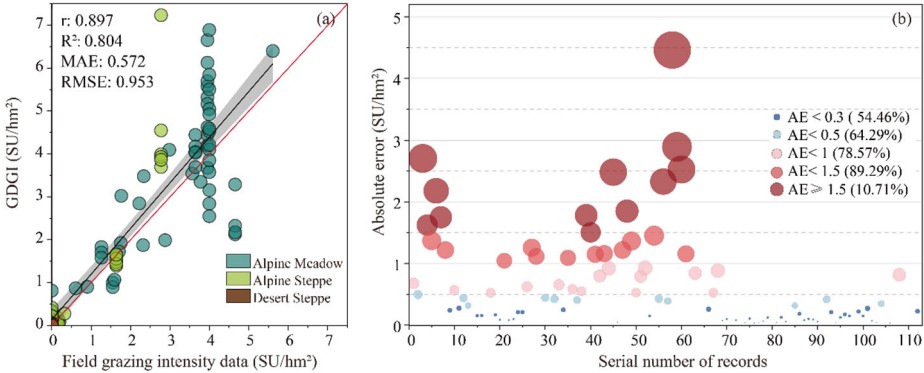

Figure 6. Validation of the GDGI dataset using 112 field grazing intensity records at the pixel scale: (a) linear fitting results; (b) absolute error (AE) distribution.

We further validated the precision of the GDGI dataset using the township-level livestock census data. Encouragingly, the evaluation results showed that the GDGI dataset has excellent performance at the township scale (Figure 7a), with $R^2$ of 0.867, MAE of 0.208 SU/hm$^2$, and RMSE of 0.276 SU/hm$^2$. In addition, similarly to the census data, the GDGI dataset indicated that some townships with few grasslands are still under high grazing pressure (Figure 7b and 7c).

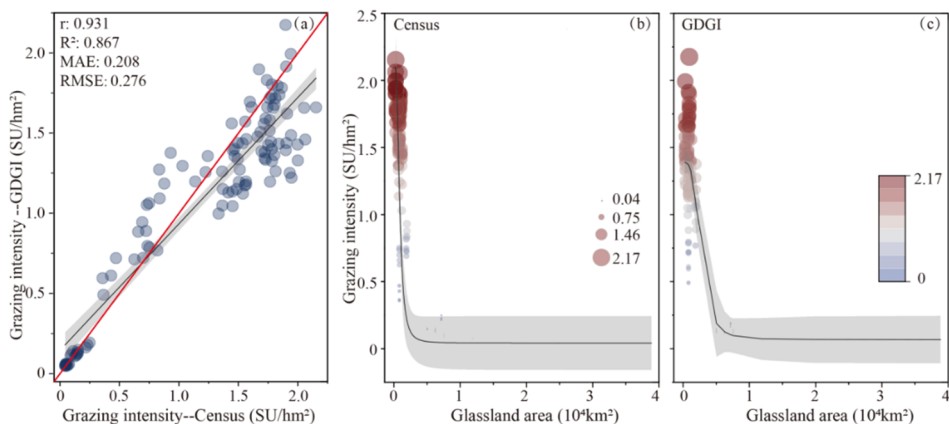

Figure 7. Validation of the GDGI dataset using census livestock data at the township level: (a) linear fit of predicted number and census data; (b-c) logistic fit of grazing intensity data and grassland area.

## 3.4 Spatio-temporal variations of grazing intensity

In terms of the temporal trends of grazing intensity, the GDGI dataset overall exhibited consistent trends with the livestock census data (Figure 8d-8f). Specifically, the census data indicated the livestock numbers remained high and largely stable from 1990 to 1997, followed by a sharp decline from 1997 to 2001, and then remained a period of fluctuation post-2001, which was successfully captured by the GDGI dataset. Moreover, the spatial heterogeneity of grazing intensity within the counties over the QTP was also effectively reflected by the GDGI dataset, a characteristic not illustrated by the census dataset. For example, areas of high grazing intensity were concentrated in the northeastern and south-central regions of the plateau, mainly including the eastern part of Qinghai Province, the southwestern part of Gansu Province, the northwestern part of Sichuan Province, and the eastern region of the Tibet Autonomous Region (Figure 8e and 8f).

Over the past 31 years, 63.95% of the plateau's grassland showed a decreasing trend in grazing intensity, with 49.80% showing significant decreases, primarily located in the eastern Sanjiangyuan area and the southwestern region of the QTP (Figure 8e and 8f). Meanwhile, grazing intensity was increasing in 36.05% of the grassland, but most of them (60.16%) did not reach the level of significance and were mainly distributed in the northeastern plateau (Figure 8e and 8f).

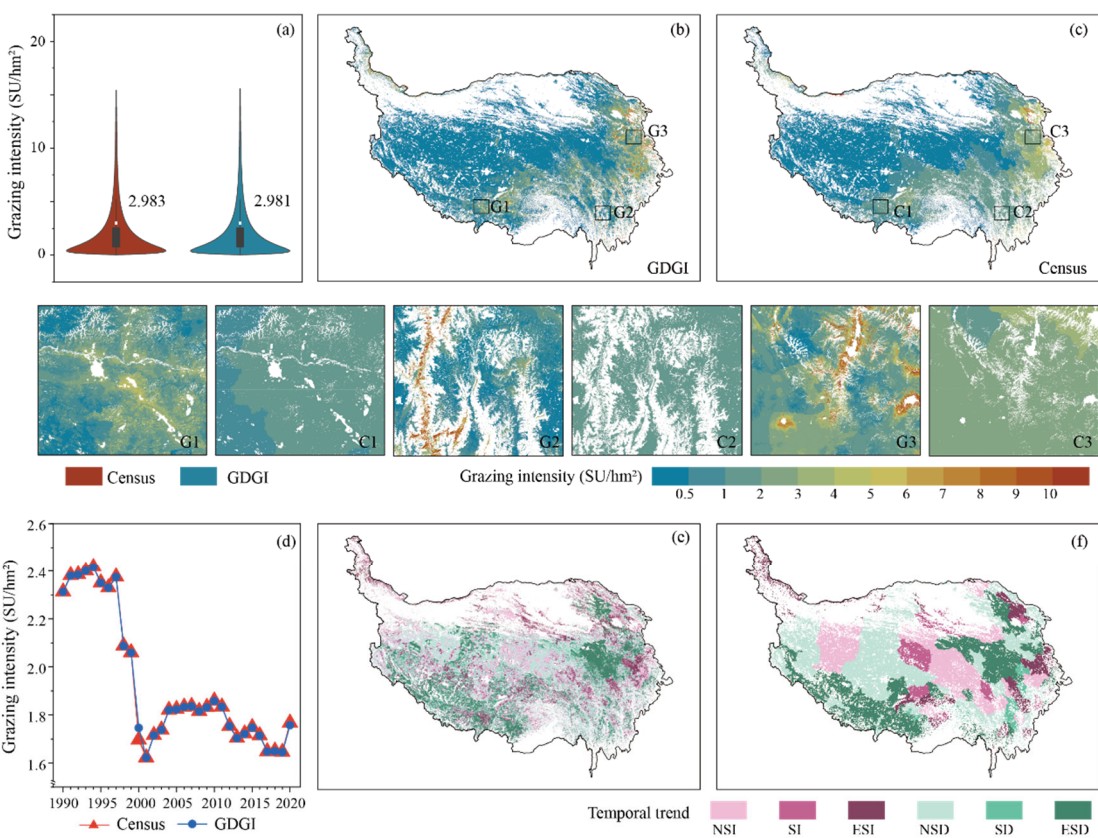

Figure 8. Validation of the GDGI maps using the census grazing data from 1990 to 2020: (a) violin plot of the census data and the predicted value; (b-c) spatial distribution in SU per pixel; (d) temporal change in SU per year (only including 124 counties with livestock census data); (d-f) spatial distribution of SU changes tested by sen's slope and Mann-Kendall. Note: ESI for Extremely Significant Increase (slope>0 & p<0.01); SI for Significant Increase (slope>0 & p<0.05); NSI for Non-significant increase (slope>0 & p>0.05); ESD for Extremely Significant Decrease (slope<0 & p<0.01); SD for Significant decrease (slope<0 & p<0.05); NSD for Non-significant decrease (slope<0 & p>0.05).

**4 Discussion**

**4.1 Comparison with other grazing intensity maps**

To further assess the effectiveness and reliability of the developed GDGI dataset, the mapping results were juxtaposed with seven publicly available grazing intensity maps covering the QTP (Table 4). It can be seen that despite their public availability, these maps lacked both in spatial and temporal resolution when juxtaposed with the GDGI maps. Our analysis was extended to four openly accessible gridded livestock datasets, including GI-Sun (Sun et al., 2021), ALCC (Liu, 2021), GI-Meng (Meng et al., 2023) and GLWs (Gilbert et al., 2018). Among the GLW series, GLW3 and GLW4 were chosen owing to their superior performances over GLW1 and GLW2, as indicated by Gilbert et al. (2018). A commonality among all five maps was the consistency for the spatial patterns of grazing intensity, with prevalent high and low intensities in the northeast and northwest regions, respectively (Figure 9). However, these maps differed significantly in terms of accuracy. As the grazing intensity maps of GLWs and ALCC were produced based on the livestock census data in 2001 and 2015, an accuracy comparison for the corresponding years was conducted among the five datasets both at county and township scale. Observations at the county scale indicate that all four datasets, with the exception of GI-Sun, are largely in alignment with the county census data (Figure 9b). When examined at the township scale, GI-Sun and GLW demonstrate the most significant discrepancies, with MRE surpassing 68%. ALCC and GI-Meng follow, recording MREs of 30.69% and 38.80%, respectively. Additionally, the GDGI shows the highest degree of accuracy in relation to the township census data, as indicated by the lowest MAE and RMSE values (Figure 9c). Moreover, the GDGI dataset spanning 31 years (1990-2020) earmarked it as a more suitable choice for long-term studies in comparison to the other four datasets. Regarding spatial distribution, the overall patterns of these grazing maps are largely consistent, exhibiting higher density patterns in the southeast and lower in the northwest. However, notable discrepancies are still apparent in the finer details. In general, in terms of visually representing the spatial distribution of livestock, the GDGI maps exhibit the best performance.

The above advantageous of the GDGI dataset are understandable. First, the livestock census data used in GDGI is more detailed, aiding in enhancing the accuracy of the estimation results. Specifically, GI-sun, ALCC, GI-Meng and GDGI all use county-level livestock statistics to map grazing intensity, whereas GLW3 and GLW4 are based on provincial-level census data to map, which results in their accuracy lagging significantly behind the four other datasets (Nicolas et al., 2016; Sun et al., 2021). Second, grazing densities are estimated by dividing the number of livestock from the statistical data, after a mask excluding theoretical unsuitable grazing areas. However, these maps differ in their definitions of suitable grazing areas. In this study, as with the GI-sun and GI-Meng maps, we considered grazing to occur only on grasslands, and further excluded unsuitable areas such as high elevations and steep slopes. This kind of definition is clearly more reasonable than the GLW series, which removed only water bodies, urban core areas, and protected areas with relatively tight regulations of human activity (Mcsherry and Ritchie, 2013; He et al., 2022). However, the GI-Meng dataset considers the core areas of protected areas as grazing-free region, it does not match the actual situation on the QTP (Jiang et al., 2023; Li et al., 2022b; Zhao et al., 2020). Those different thresholds for the definition of suitable grazing areas are account for the fact each map has different theoretical grazing regions. Third, these maps decompose the livestock census data to pixels based on different mathematical theories, which also leads to differences in prediction accuracy across maps. Specifically,

ALCC used a multivariate linear regression algorithm to predict grazing intensity, which has been
shown to be significantly inferior to the RF machine learning method employed by GI-Meng, GLW3
and GLW4 (Nicolas et al., 2016; Li et al., 2021). In this study, we used the ET model to predict
livestock numbers and achieved higher accuracy accordingly. Finally, differences in the selection of
factors affecting livestock distribution across maps may also lead to differences in map accuracy.
Specifically, GI-sun only used the NPP as indicator, but it is not simply linearly related to grazing
intensity (Sun et al., 2021; Ma et al., 2022; Gilbert et al., 2018). ALCC considered the population
density, NPP, and terrain as indicators, which are also incomplete considerations of the influencing
factors. On the other hand, GLW series dataset considered 12 factors, such as NDVI, EVI, population
distribution and elevation. GI-Meng dataset incorporated 14 factors including NDVI, soil PH, available
nitrogen, available phosphorus, and available potassium. However, GLWs and GI-Meng ignored the
decrease in the prediction accuracy due to redundancy among the factors. In this study, we selected
factors related to grazing activities including terrain, climate, environment and social factor, and
constructed a prediction model with seven factors including population density, elevation, climate, and
HNPP. Unlike other livestock products, this study used HNPP for the first time to replace the
commonly used NPP, or NDVI, or EVI as indicator, which has be proved to be more accurately
expressed the relationship between livestock and grassland (Huang et al., 2022).

Table 4. Summary of map-derived parameters for this study and other seven public gridded livestock datasets covering the QTP.

| Dataset | Accessibility | Census | Temporal resolution | Spatial resolution | Period (years) | Method | Livestock type |
|---------|---------------|--------|---------------------|--------------------|----------------|--------|----------------|
| GDGI | Yes | County | annual | 100 m | 1990-2020 (31) | ET | Standard SU |
| GLW3 | Yes | Province/sub-Province | annual | 0.083°($\approx$10 km) | 2001 (1) | RF | Cattle, ducks, pigs, chickens, |
| GLW4 | Yes | Province/sub-Province | annual | 0.083°($\approx$10 km) | 2015 (1) | RF | sheep, goats |
| GI-Sun | Yes | County | five-year interval | 1 km | 1990-2015 (6) | LRA | Standard SU |
| ALCC | Yes | Province/sub-Province | annual | 250 m | 2000-2019 (20) | MLR | Standard SU |
| GI-Meng | Yes | County | annual | 0.083°($\approx$10 km) | 1982-2015 (34) | RF | Standard SU |
| GI-Li | No | County | five-year interval | 1 km | 2000-2015 (4) | DNN | Cattle and sheep |
| GI-Zhan | No | County | season | 15″ ($\approx$500 m) | 2020 (2) | RF | Standard SU |

438    Note: LRA is the abbreviation of linear regression analysis.

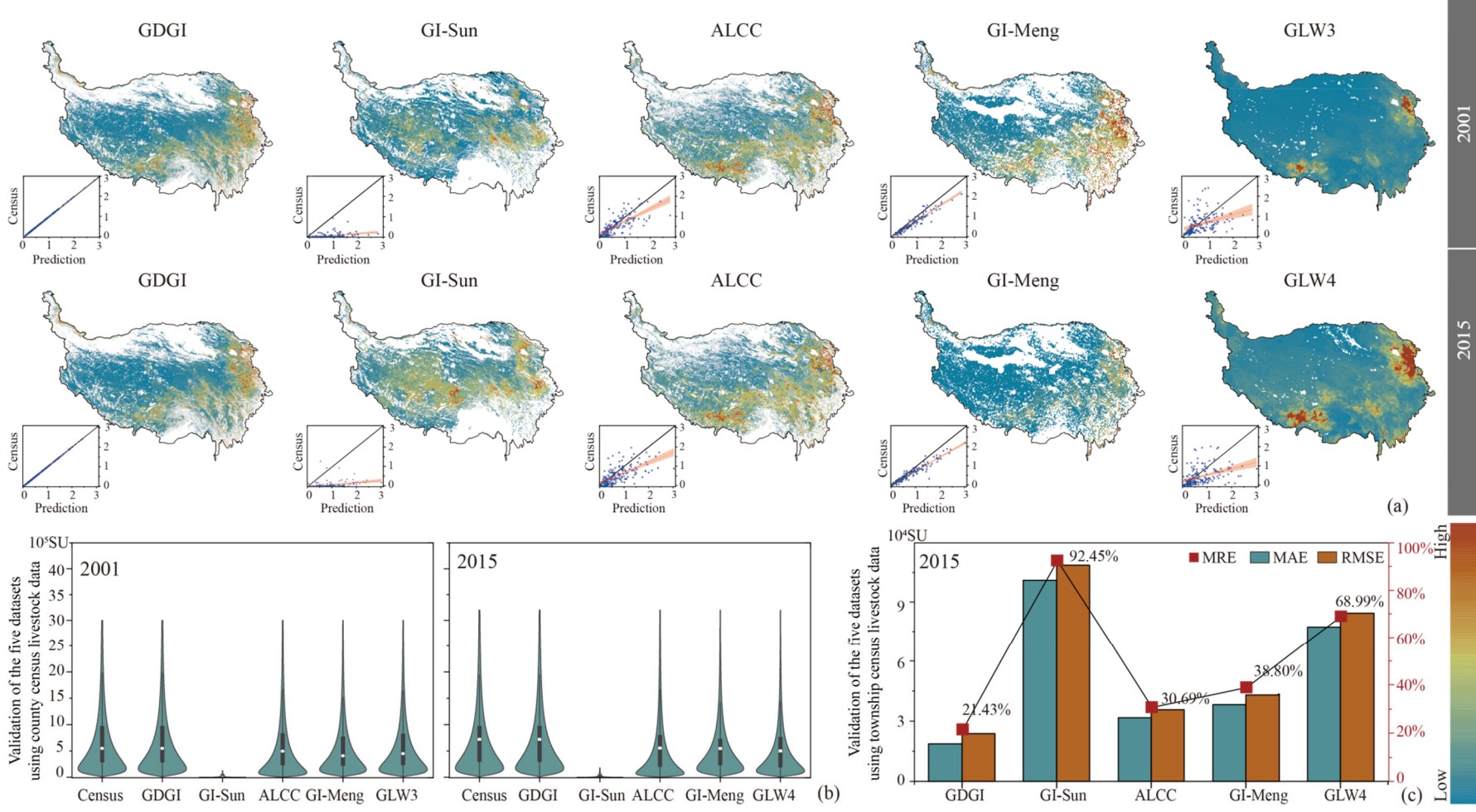

Figure 9. Comparisons of different grazing datasets for the years 2001 and 2015: (a) spatial patterns; (b) predicted livestock number and census data at county scale; (c) accuracy evaluation between predicted livestock number and census data at township scale.

## 4.2 Spatial heterogeneity of grazing intensities

In general, the multiyear average grazing intensity on the QTP increased from west to east during 1990 to 2020, with broad spatial heterogeneity (Figure 8). Highest grazing intensity was found mainly in the northeastern and south-central regions of the Plateau (mostly higher than 5.0 SU/hm²), while they were lowest in the northwest (mostly less than 1.0 SU/hm²). Over the past 31 years, the average grazing intensity decreased across most of the Plateau, but 36.05% of the entire QTP grassland still encountered continuous grazing intensity increase, especially in the northeastern regions (Figure 8).

The spatial heterogeneity of grazing intensities on the QTP may be attributed to the following reasons. First, complex geographic and climatic conditions on the QTP determine the heterogeneity of grassland, which in turn affects livestock distribution (Wang et al., 2018; Wei et al., 2022). In general, the grazing intensity patterns shown in the GDGI maps are basically consistent with the stocking rate threshold patterns in the QTP grasslands, both decreased from east to west (Zhu et al., 2023a). This phenomenon partially reflects the heterogeneity of the grasslands, as the alpine meadows and the steppes mainly distributed in the east and the west, respectively. Second, the dynamics of socio-economic development are obviously another important factors determining grazing intensity. In areas falling behind in terms of the socio-economic indicators, herders prefer to increase livestock in efforts to improve household incomes, leading to greater pressure on grasslands in these regions (Fang and Wu, 2022). In addition, the perceived increases in human population also resulted in the considerably increased need to more livestock (Wei et al., 2022).

The grazing intensity dynamics across the QTP are partly reflective of the impacts of various management policies that have been implemented over distinct periods. For example, a significant increase in grazing intensity on the QTP was observed in the early 1990s, potentially a direct result of the introduction of the household contract responsibility system. Moreover, the grazing intensity experienced a pronounced decline from 1997 to 2001, as illustrated in Figure 8d, indicative of the adverse effects of natural disasters. Notably, the severe snowstorms that struck Naqu in the central QTP during 1997-1998 are documented to have caused the mortality of over 820,000 livestock (Ye et al., 2020). Figure 8d further delineates a considerable upsurge in grazing intensity on the QTP between 2000 and 2010, aligning with the trends reported by Sun et al. (2021) and Li et al. (2021). This observed increase may be attributed to a rebound in grazing activity following the aforementioned natural disasters. In addition, Figure 8d indicates a sustained decrease in grazing intensity post-2010 across the plateau, which can be predominantly ascribed to the implementation of extensive ecological conservation projects.

## 4.3 Implications for grazing management

Nearly half of the grasslands on the QTP have been reported to be degraded over the past four decades (Wang et al., 2018; Dong et al., 2020), with some reports even indicating that the degraded grassland has reached 90% (Wang et al., 2021). It is widely recognized that overgrazing is the predominant and most pervasive unsustainable human activity continuing to drive grassland degradation on the QTP (Wang et al., 2018; Chen et al., 2019). Generally, these degraded grassland on the QTP can be effectively restored by adaptive management (Wang et al., 2022). However, better management of grasslands requires a deeper understanding of the anthropogenic activities, which still remain an important challenge and can be effectively addressed by the GDGI dataset.

According to the GDGI maps generated in this study, high-intensity grazing activities are mainly concentrated in the northeastern as well as the south-central part of the QTP, with the grazing intensity in some areas even nearly more than ten times than the average value of the entire plateau (Figure 6b), and have exceeded the stocking rate threshold of these grasslands (Zhu et al., 2023a). Population growth and the related increasing livelihood demands is one of the main reasons for this increase. To meet daily needs and enhance household income, the herders have endeavored to increase livestock, thereby intensifying grazing pressures on the grasslands over the QTP (Fang and Wu, 2022; Abu Hammad and Tumeizi, 2012). Although the current average grazing intensity in the northwest QTP (around 1.0 SU/hm²) is below their average stocking rate threshold (around 1.5 SU/hm²) (Zhu et al., 2023a), the grassland management should still be given adequate attention. Because as the most arid areas with low stocking rate threshold on the QTP, the grazing intensity in this region has been increasing in recent years. Nevertheless, it must be noted that the stocking rate threshold may exceed the carrying capacity, because it is predicted to lead to an extreme grassland degradation (Zhu et al., 2023a). The GDGI dataset also showed a similar pattern between the grazing intensity data and the WorldPop data near the built-up areas, indicating higher grazing intensity around settlements than other regions on the QTP. In addition, the GDGI dataset also indicate that from 1990 to 2020, although the grazing intensity of the Plateau has generally decreased, the hotspot areas for grazing activities have remained almost unchanged. This implies that these regions should be the focus of adaptive grassland management to effectively prevent grassland degradation, mainly based on the grass–livestock balance which varies by time and space.

Encouragingly, the GDGI dataset show that the grazing intensity for two-thirds of the entire QTP grassland decreased over the past 31 years, which is also consistent with other studies (Sun et al., 2021; Li et al., 2021). Recent decades of biodiversity protection, active restoration projects as well as management measures, such as nature reserves, grazing exclusion, part grazing ban combined with fencing enclosure, are believed to have driven these decrease (Deng et al., 2017; Li and Bennett, 2019). In addition, most grassland in the eastern Sanjiangyuan, the mid-eastern Changtang, and the northern foothills of the Himalayas, showed a significant decrease with grazing intensity (Figure 6e), indicating the importance of protected areas on preventing overstock and grassland degradation. Meanwhile, the GDGI maps also show that the grazing density varies greatly among protected areas, possibly owing to the difference in policy implementation. For instance, it can be seen from the GDGI maps that grazing intensity are increasing in some protected areas, especially several wetland nature reserves on the Zoige plateau (Figure 6e). Moreover, the average grazing intensity in all nature reserves on the QTP has overall increased from 1990 to 2020, although their increase rate is much lower than the non-protected areas (0.0125 SU/hm²·10a vs 0.0304 SU/hm²·10a), which implies that grassland management in protected areas still needs to be strengthened on the QTP.

The grazing initiatives in alignment with the Sustainable Development Goals (SDGs) on the QTP can benefit from the GDGI dataset. Firstly, determination a reasonable stocking rate is vital to prevent overstocking of the pastures, which will possibly induce extreme grassland degradation (Zhu et al., 2023a). Stocking rate determination can be optimized by using our grazing intensity maps and the stocking rate threshold maps of the QTP. Secondly, the GDGI maps can contribute to strategic placement of fence, which is a common practice adopted to prevent grassland degradation on the QTP. Building fences in areas with high grazing intensity and exceeding the carrying capacity can improve the effectiveness of fence construction (Zhou et al., 2023; Zhang et al., 2023). Thirdly, the GDGI

dataset can provide a solid support for promoting effective nature reserve management, which in total covering nearly one third of the entire QTP. For example, the GDGI maps showed that grazing activities still exist in most nature reserves on the Plateau, although most of them have significantly lower grazing intensities compared with their adjacent non-protected areas. By using the GDGI maps, the conflict between ecological protection and grazing activities in nature reserves can be alleviated. Finally, our grazing intensity maps can act as a basic dataset to support other grassland-related policies. Currently, these policies on the QTP often adopt a one-size-fits-all approach to determine the carrying capacity and carry out ecological compensation, which may lead to overstock or unfair financial distribution (Wang et al., 2022). The grassland management strategies balancing carrying capacity and stocking rates are more likely to result in optimal management choices for policymakers and stakeholders, and our GDGI maps can contribute to this decision-making processes.

## 4.4 Uncertainties and limitations

Although this study has collected as reliable datasets as possible, users of the GDGI products should be cognizant of inherent uncertainties and limitations within these datasets. Notably, the mean relative error of the GDGI dataset spanning 1990 to 2020 was recorded at 4.2% (Figure 4a), calculated from the average errors across 182 counties within the QTP that had accessible livestock census data. Furthermore, approximately 8.26% of grassland areas exhibited a relative error exceeding 1.0 SU/hm$^2$ (Figure 4b). Such discrepancies arise from several limitations that were subsequently propagated to the final grazing intensity maps, thereby contributing to the dataset's overall uncertainties.

Firstly, the estimations of grazing intensities were fundamentally conservative, primarily due to the lack of comprehensive input data. Livestock numbers, derived from year-end data at the county level, inadvertently led to underestimations of grazing intensity by not accounting for livestock off-take rates. Likewise, the evaluation focused solely on livestock grazing intensity, excluding wild herbivores and forage-dependent livestock, which potentially underestimate actual grazing pressures on the QTP. Additionally, despite identifying seven main factors influencing livestock distribution, the study did not encompass all potential factors, such as fencing, forage availability, road proximity, and season transformation in grazing practices. Moreover, to align with county-scale livestock census data, we averaged the environmental factors at the county-scale. Although this approach have been widely used on the hypothesis that a consistent causal relationship between livestock intensity and environmental factors persists across various scales (Robinson et al., 2014; Nicolas et al., 2016; Li et al., 2021; Meng et al., 2023), it might oversimplify the intricate dynamics between grazing intensity and lead to a certain degree of estimation inaccuracies. In addition, the reliance on linear extrapolation to Supplementary missing gridded 100-m population density data from 1990-1999 introduced further uncertainties due to the limited resolution (1-km) and interval (5-year) of the ChinaPop dataset.

Secondly, the modeling process for mapping grazing intensity also suffered from several challenges. For instance, the ET model was trained with a limited sample size of 4,998 and applied to a vast area consisting of 150 million pixels, which could compromise the model's accuracy. In addition, despite the ET model's design to reduce overfitting risks by using randomly selected features and partition decision, the potential for overfit effects still remained, particularly when faced with a high number of output classes or insufficient sample sizes (Geurts et al., 2006; Galelli and Castelletti, 2013). In fact, this limitation was evident in this study, as the generalization capability of the ET model was restricted by the disparity between the number of training samples and the total number of pixels, leading to predictions that often exceeded actual livestock census (Figure 4a).

Thirdly, our methodological framework for high-resolution gridded grazing dataset mapping was developed based on the assumption that all grassland were accessible to livestock. However, in reality, the amount of available grassland was less due to fencing and grazing bans on the QTP (Zhan et al., 2023). Moreover, transhumant herders generally follow a seasonal calendar for summer pastures and winter pastures on the QTP. However, we did not consider this seasonal movements due to data limitations, which further restrict the analysis of seasonal livestock distribution patterns (Kolluru et al., 2023). Additionally, the model's reliance on human population as a proxy for livestock locations overlooked the possibility of high grazing intensity in areas with low human populations on the QTP, particularly in regions designated for summer pastures.

Finally, it is important to note that gathering livestock census data in the Qinghai-Tibet Plateau presents significant challenges, leading to a scarcity of livestock validation data in this study, particularly at the township and pixel scales. This limitation may, to some extent, impact the reliability of the grazing intensity data we have presented.

In summary, all these limitations associated with input data, the modeling process, and the methodological framework collectively contribute to the uncertainties and potentially reduce accuracy of the GDGI maps. We henceforth recommend that future research should aim to incorporate more detailed data, consider additional influential factors, enhance key dataset's time-series consistency, and refine the methodological framework to improve the accuracy of grazing intensity mapping.

**5 Data availability**

The annual gridded grazing intensity maps of the QTP spanning from 1990 to 2020 are accessible at the following link: https://doi.org/10.5281/zenodo.13141090 (Zhou et al., 2024). Each map is catalogued by year and recorded in GeoTIFF format, with values represented in $SU/hm^2$ per year. These datasets, with a spatial resolution of 100 m and annual temporal resolution, utilize the WGS-1984-Albers geographic coordinate system. To streamline data transfer and download processes, the comprehensive 31-year dataset has been compressed into a ZIP file, readily available for download and compatible with Geographic Information System (GIS) software for viewing.

**6 Conclusions**

In this study, we introduce a framework utilizing ET machine learning algorithms to achieve fine-scale livestock spatialization, subsequently generating the GDGI dataset across the QTP. The GDGI has a spatial resolution of 100 m and expands 31 years from 1990 to 2020. It is consistent with livestock census data of the QTP, and has a relatively higher precision than previous datasets with MAE of 0.006 $SU/hm^2$ based on 4,998 independent test samples. In addition, the accuracy evaluations at both pixel-level and township-level underscore the outstanding reliability and applicability of the GDGI dataset, which can successfully capture the spatial heterogeneity and variation in grazing intensities in greater details. Moreover, comparisons between the GDGI dataset and other existing grazing map products further proved the robust and efficient of our dataset, and demonstrate the validity of the proposed framework in the research of livestock spatialization. The GDGI dataset presented in this study can address existing limitations and enhance the understanding of grazing activities on the QTP. This, in turn, can aid in the rational utilization of grasslands and facilitate the implementation of informed and sustainable management practices.

**Supplementary.**

For gridded datasets influencing grazing that are not directly available, or that do not meet
spatio-temporal resolution requirements—such as those pertaining to population density, temperature,
precipitation, and HNPP—we have delineated the processing or creation procedures in the
Supplementary file.

**Author contributions.**

615 T.L. conceived the research; J.Z. and J.N. performed the analyses and wrote the first draft of the
616 paper; N.W. and T.L. reviewed and edited the paper before submission. All authors made substantial
contributions to the discussion of content.

**Competing interests.**

The authors declare that they have no conflict of interest.

**Acknowledgements.**

We would like to thank the Bureau of Statistics of each county over the QTP for providing the
census livestock data.

**Financial support.**

This research was supported by the Second Tibetan Plateau Scientific Expedition and Research
Program (STEP), Ministry of Science and Technology of the People's Republic of China (grant no.
2019QZKK0402) and the National Natural Science Foundation of China (grant no. 42071238).

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
