# Peer review of "Annual high-resolution grazing intensity maps on the Qinghai-Tibet Plateau from 1990 to 2020"

_Earth System Science Data, 2023_

## Author Comment (AC2)

[Figure]

Figure R1. Comparisons of the reconstructed population density maps and the WorldPop dataset in 2000.

[Figure]

Figure R2 Comparisons between the WorldPop and the ChinaPop dataset in 2015.

[Figure]

Figure R3. Comparisons for the distribution patterns of grazing intensity maps, environmental variables, and census livestock data.

[Figure]

Figure R4. Comparisons of the accuracy by different downscaling methods in predicting NPP distribution.

---

## Author Response (AR1)

**Response to CC1**

**General Comments:**

The high-resolution grazing maps are of significant important to researchers and grassland managers, as they substantially improve our understanding of the threats posed to grasslands by overgrazing. Moreover, these maps have potential applications in various areas. I think this study makes very useful contribution to the field, the maps provided are the first of their kind for the Tibetan Plateau, making them a crucial resource for studying grazing intensities in this region. As such, their publication is highly recommended. Overall, the study's novelty is appreciable, and its quality aligns with the standards of ESSD. However, the manuscript will need some minor revisions before it can be accepted for publication.

**Response: Thank you very much for your overall positive words about our paper. We are delighted to hear that our manuscript received such feedback. We have improved the manuscript according to you and another reviewer's suggestions. In the following sections, you will find our detail responses to each of your points and suggestions. We are grateful for the time and energy you expended on our behalf.**

**Specific Comments:**

Line 48: There are a number of typos need to be carefully revised in the references. For example, it is unclear whether there should be a comma after 'et al.' in the author citations.

**Response: Thanks for raising this point. We have checked and revised all the references one by one in the revised manuscript with reference to the requirements of ESSD journal (see through the text).**

Line 94: The text incorrectly states thirteen factors, whereas only four are discussed. This discrepancy should be corrected.

**Response: We have reconfirmed that Li et al. (2021) did select 13 environmental factors from four aspects: land use practice, topography, climate, and socioeconomic. In the revised version, we have modified the vague expressions (see L85~88).**

Lines 142-143: I think although this study makes important contributions to the grazing intensity mapping, the methodological framework is generally not new. This needs to be addressed if the manuscript is to be revised.

**Response: Thanks for raising this point. We have deleted the word of "novel" in the revised version (see L138).**

Line 166: Figure 1 lacks a compass.

**Response: Since the WGS_1984_Albers projection was used for Figure 1, placing a compass would contradict the direction indicated by the latitude lines, so the direction was indicated by the latitude and longitude lines in Figure 1.**

Line 176: A comma is needed in the number 4998 for consistency with standard numerical formatting.

**Response: Thanks for raising this point. We have corrected these numbers for consistency with standard numerical formatting in the revised manuscript (see through the text).**

Lines 228-231: The descriptions of the methodology, particularly the specific machine learning algorithms used is poor and needs to be improved.

**Response: Thank you for this great suggestion. We totally agreed with you that the methodology section need to be improved. In light of your feedback, we have introduced more detail information about the five machine learning algorithms in the appendix (see L148~182 in the supplement file for details).**

Lines 382-415: This section would benefit from referencing additional literature related to grassland management on the Tibetan Plateau. For instance, including Wang et al. (2023) published in Nature Reviews Earth & Environment could provide more insights into the novelty and contribution of the GDGI dataset.

**Response: We appreciate this valuable comment. In the revised version, we have had an in-depth discussion related to grassland management in the Tibetan Plateau by referencing more literatures, including the paper you mentioned (see L425~487). We hope you find these revisions rise to your expectations.**

**Response to RC1**

**General Comments:**

Livestock is a crucial contributor to global food systems through the provision of essential animal proteins and fats, and plays a significant role in supporting human survival and socio-economic development. This study developed a long-term and high-resolution grazing intensity dataset in the Qinghai-Tibet Plateau from 1990 to 2020 by integrating machine learning algorithms, census data, and multiple environmental variables and socio-economic factors. The manuscript is well structured, but the discussion section needs to be improved. The machine learning model performs well. The spatial pattern of the grazing intensity map is also consistent with previous studies and looks more reasonable. The dataset is valuable for the research focusing on climate change, human activities, and their interactions with ecosystem dynamics in the Qinghai-Tibet Plateau. However, I have some concerns in the data preparation, model training and validation, which are provided blow.

**Response: Thank you for these overall positive words. We have thoroughly addressed all of your comments and suggestions, especially for your concerns in the data preparation, model training and validation, as well as the discussion. Please see our detail responses to the individual comments in the following sections. We hope that our revisions rise to your expectations.**

**Major comments**

1) The first major concern is the input data for spatialization. The key input datasets for the grazing intensity spatialization are population density, climate data, terrain, and HNPP.

Population density data:

 Line 66-68 in supplementary materials. WorldPop data at 100 m resolution from 2000 to 2020 and Population at 1km resolution from 2000 to 2015 was used to reconstructed the population at 100 m during 1990 to 2000. For each pixel, only four records were used to fit the linear model, there is large uncertainty using such as model in estimating the population in 1990 and 1995. The downscaling method or bias correction of population data (1990-2000) is not reasonable and can be improved, check the bias correction method from He et al. (2022). He, S., Zhang, Y., Ma, N., Tian, J., Kong, D., and Liu, C.: A daily and 500 m coupled evapotranspiration and gross primary production product across China during 2000–2020, Earth Syst. Sci. Data, 14, 5463–5488, https://doi.org/10.5194/essd-14-5463-2022, 2022.

**Response: Thanks for raising this important point. We totally agree with you that fitting a linear model using only four pairs of records for each pixel may introduce uncertainty. Nevertheless, we have no other choice because the ChinaPop dataset at 1-km resolution is at 5-year intervals. In light of your feedback, we address the uncertainty that this shortcoming may introduce in the revised discussion section (see L488~533). As for the bilinear interpolation and delta change method you mentioned, which were used in the He et al. (2022) article, we have actually paid attention to it before. The reason why they were not adopted as the downscaling and bias correction method in our manuscript is mainly based on the following reasons:**

**Firstly, as the bilinear interpolation method automatically generates new intermediate values during the interpolation process, it is very suitable for indicators with continuous values, like surface temperature in the He et al. (2022) article. It is obvious that population density is not a continuous variable. As for the nearest neighbor algorithm, it does not change any pixel values in the resampling process, thus avoiding the unanticipated bias caused by pixel value changes, and is suitable for indicators with discontinuous values. Therefore, in this study, the nearest neighbor method was used instead of bilinear interpolation method for downscaling process.**

**Secondly, another reason we used the nearest neighbor algorithm is to be consistent with the WorldPop dataset, which also used the nearest neighbor method for their resampling process.**

[Figure]

Figure R1. Comparisons of the reconstructed population density maps and the WorldPop dataset in 2000.

**Thirdly, since the 100-m WorldPop dataset is one of the basic data sources in this study, the criterion for determining the effectiveness of the downscaling and bias correction method, should be based on the compliance of the simulation results with the WorldPop dataset. In other words, the more similar the reconstructed data and the WorldPop dataset, the better performance of the methods are considered to be. Taking the population density map in 2000 as an example, we used the bilinear interpolation method to resample the 1-km population density data to 100 m, and then corrected the population distribution data with Delta change method. In addition, we also used the nearest neighbor and liner regression methods to resample the 1-km population density data. Last, we compared these two 100-m population density data with the WorldPop dataset. It can be seen that the reconstructed population density data by using the nearest neighbor and liner regression methods are more consistent with the WorldPop data (see Figure R1 in the attached response file).**

- Moreover, the WorldPop dataset is not reliable in the QTP. On the contrast, the 1km population from obtained from the Resource and Environment Science and Data center of the Chinese Academy of is more accurate, because it is developed by using the county-level census data.

**Response: This is another good point. In this study, the reason we chose the 100-m WorldPop dataset instead of the 1-km ChinaPop dataset are mainly based on the following reasons:**

**First, the WorldPop dataset has higher spatial and temporal resolution (100-m and annual), which provides important support for this study to eventually provide annual grazing dataset at 100-m resolution. While the ChinaPop dataset has a spatial resolution of 1 km and a temporal resolution of five years, which can not serve this study well.**

**Second, the WorldPop dataset is actually more accurate than the ChinaPop dataset on the QTP. It is true that the total population number of the ChinaPop dataset does better match the actual statistics than the WorldPop dataset, because it was developed based on county-level census data. However, the spatial heterogeneity of the ChinaPop dataset is coarse, whereas the WorldPop dataset is able to show more details (Figure R2a,2b in the attached response file). For example, the ChinaPop dataset has populations in lakes and glaciers, which is obviously unreasonable. Moreover, the WorldPop dataset was able to successfully identify the uninhabited areas in the northwestern part of the plateau, whereas the ChinaPop dataset can not (Figure R2a,2b in the attached response file). In addition, the WorldPop dataset can identify 204 urban areas on the QTP, while the ChinaPop dataset can only identify 116 urban areas with fuzzy boundaries (Figure R2c in the attached response file).**

**Third, other scholars have also shown that the WorldPop dataset is more accurate than the ChinaPop dataset on the Tibetan Plateau (Li et al., 2020). Li, L.**

H., Zhang, Y. L., Liu, L. S., Wang, Z. F., Zhang, H. M., Li, S. C., and Ding, M. J.: Mapping Changing Population Distribution on the Qinghai–Tibet Plateau since 2000 with Multi-Temporal Remote Sensing and Point-of-Interest Data, Remote. Sens., 12, 4059, https://doi.org/10.3390/rs12244059, 2020.

[Figure]

Figure R2 Comparisons between the WorldPop and the ChinaPop dataset

Besides, there are only 7 snapshots for population after data harmonization. How did you get the population density data for rest years to train the machine learn models and conduct the grazing intensity spatialization?

**Response: Thanks for raising this important point. After obtaining population density maps for 1990 and 1995, we generated maps for the remaining years in 1990-1999 by linear extrapolation. We have added this information to ensure clarity according to your comments in the revised version (see L34~37 in the supplement file for details).**

I did check the grazing intensity dataset and found that pattern of grazing intensity near the urban/built-up area was quite similar with WorldPop data. The influence of population density also resulted in some abrupt changes in the southeast of Qinghai Lake.

**Response: Many thanks for your comment. In this study, we assumed that grazing activities are confined solely to grassland, thus have excluded urban areas. Nevertheless, livestock grazing activities can still be influenced by a variety of social and environmental factors, which also including population distribution. Therefore, it is reasonable for grazing intensity patterns to be similar with the WorldPop data. In fact, other studies also showed that grasslands closer to settlements face more degradation, because of more grazing intensities. Li, C. X., de Jong, R., Schmid, B., Wulf, H. &Schaepman, M. E. Spatial variation of human influences on grassland biomass on the Qinghai–Tibetan Plateau. Sci. Total Environ. 665, 678–689 (2019). https://doi.org/10.1016/j.scitotenv.2019.01.321.**

**In fact, the pattern of grazing intensity in this study are not only similar with population data, but also with other factors used in this study. Taking the**

**population density and slope as example, it can be seen that the grazing intensity pattern is not only similar with the population density pattern, but also similar with the slope pattern (Figure R3a, 3c, 3d in the attached response file).**

**It is true that there is an abrupt change in grazing intensity in the southeastern part of the Qinghai Lake in our GDGI map. In fact, this pattern is consistent with the real condition in this region, as the census data has showed that several counties in the southeastern part of Qinghai Lake have high livestock numbers and relatively low grasslands, which in turn leads to higher grazing intensity (Figure R3b in the attached response file). In addition, this pattern can also be proved by other grazing intensity maps (see Figure 8 in the main text).**

[Figure]

Figure R3. Comparisons for the distribution patterns of grazing intensity maps, environmental variables, and census livestock data.

HNPP data:

- MODIS terra was launched in 1999, it is impossible to get the NPP product before 2000. How did you the MOD17A3 NPP data from 1990 to 2000. Please check the data source carefully.

**Response: Thank you for pointing out this error. We sorry about this mistake. In this study, the NPP data for the 2000-2020 period (NPP-I) is obtained from the Land Processes Distributed Active Archive Center (https://lpdaac.usgs.gov), and the NPP data for 1990-2015 (NPP-II) was obtained from the Global Change Research Data Publishing and Repository (http://www.geodoi.ac.cn). Chen Pengfei. Monthly NPP Dataset Covering China's Terrestrial Ecosystems at North**

**of 18°N (1985-2015), 2019, 3(1): 34-41. DOI: 10.3974/geodp.2019.01.05. In the revised manuscript, we have corrected this error (see L119~120 in the supplement file for details).**

The downscaling or bias correction method of NPP data (1990-2000), same as population density data, is not reasonable and can be improved.

**Response: We appreciate this valuable comment. The reason we did not downscale the NPP data using a combination of bilinear interpolation and Delta change correction is similar as the population density downscaling process. Taking the year of 2000 as an example, we compared the estimated NPP data obtained using bilinear interpolation and Delta change correction with the results obtained using the nearest neighbor method. It can be seen that the estimated NPP by using the nearest neighbor method are more similar with the real Modis NPP values (Figure R4 in the attached response file). The reasons for such a result are similar as the population density downscaling process above, please refer to the responses in that section.**

[Figure]

Figure R4. Comparisons of the accuracy by different downscaling methods in predicting NPP distribution.

The Thornthwaite Memorial model to estimate the potential NPP in QTP, is it suitable for QTP? I suggest add several figures to show the spatial pattern of HNPP and reflect the contribution of human activities on grassland in the QTP. How is the relationship between HNPP with population density?

**Response: Thanks for raising this important point. Our previous literature statistics have showed that there are at least more than twenty papers using the**

**Thornthwaite Memorial model to estimate PNPP on the Tibetan Plateau after 2010. Therefore, it is reasonable to believe that this model is appropriate for the QTP. In addition, based on your suggestion, we have added one map to show the spatial pattern of HNPP (see Figure S2 in the supplement file for details). As for the relationship between HNPP and population density, we calculated the relationship between them by using the Bivariate Moran's I index. The results showed that Moran's I was 0.2, which indicate there is positive relationship between population density and HNPP . To verify the reliability of the results, we further performed randomization with 999 permutations, which showed a p-value of 0.001 and a Z-score of >2.58. This indicates that there is a significant positive correlation between population density and HNPP. This means that where population density is high, HNPP are also high (see Figure S2 in the supplement file for details).**

2) Another major concern is the spatialization model. Because the machine learning models were trained and validated by using county-level data, it should be very careful before applying to the prediction at grid scale. Specifically, the data you use to train the model may not be representative all the grids because it only reflects an average condition of each county. If the county-level model was applied to grid scale, there would be some bias in the spatial prediction though the model results look good at county-level, such as low estimation at high grazing intensity area and high estimation at low grazing intensity area. Thus, a statistical distribution comparison between the training data and grid data should be conducted to prove the model is transferable from county-level to grid-level.

**Response: This is another important point. We totally agree with you that there may be some bias when the model trained at county-level was applied to grid scale. Therefore, a statistical distribution comparison between the training data and grid data is necessary. In fact, we do show this comparison in our manuscript (see Figure 4 in the main text). Unfortunately, we lacked the statistical analysis. In light of your feedback, we have given the statistical analysis results in the revised version (see L278~282 in the main text), which show that the model trained at county scale can be downscaled to the pixel scale. We hope you find these revisions rise to your expectations.**

3) The data validation (section 3.2 & 3.3) is quite weak. Especially, the validation in section 3.2 is not convincing and meaningless, because the grid data already corrected using the census data. Thus, the above validation is not enough because the newly grazing intensity data is grid-based and has very high resolution (100 m). Validation by using site observed grazing intensity is necessary as that in Meng et al. (2023).

**Response: Thanks for raising this important point. We totally agree with your comment and have deleted the validation analysis at county scale in Section 3.2. Moreover, we have added the validation of the GDGI dataset at the pixel scale (see L 289-298 in the main text). Also, we have provided detail information on these ground-truth validation data in the appendix (see Figure S3 and Table S3 in the supplement file for details).**

*Minor comments*

1) Line 16: "socio-economic", "social economic" should be consistent in the whole manuscript.

**Response: Done. Thank you (see L15, L189 and L413).**

2) Line 16: replace "land cover" as "vegetation index". In section 2.2, "land cover" is not included in the factors affecting grazing activities.

**Response: We apologize for the confusion caused by the description as you mentioned. In the revised version, we have replaced "land cover" as "vegetation" (see L15).**

3) Line 143: The methodological framework is not novel. Figure 2 and the four major steps in section 2.3 look quite similar with the method in Meng et al. (2023).

**Response: Many thanks for your comment. We have deleted the word of "novel", and revised the flowchart in the revised manuscript (see L137 and Figure 2 in the main text).**

4) Line 171-178: Census livestock data in some counties were not available, how did you process these counties when conduct spatialization in the missing-data year?

**Response: Thanks for raising this important point. In this study, these missing data were processed by using the following two methods. Specifically, for these counties belonging to the same prefecture, including counties in Ganzi and Aba prefectures in Sichuan Province, we used the livestock census data at the prefecture-level to carry out spatialization. For these counties in Yunnan Province, since they belong to different municipalities, it is not reasonable to replace them with municipal-level data. For these counties without livestock census data for some years, we supplemented the missing data by linear interpolation with grazing density data in available year. We have stated the process clearly in the revised manuscript (see L169~175).**

5) Line 179-182: A spatial distribution map of the towns with available livestock data could add in the supplementary materials.

**Response: Thank you for your suggestions. A spatial distribution map of the townships with available livestock data, as well as the site of ground truth validation data have been added in the revised version (see Figure S3 and Table S3 in the supplement file for details).**

6) Line 209: revise "mappingd".

**Response: Done. Thank you (see L221).**

7) Line 213: The author used the 30 m annual land cover dataset (CLCD) to derive grassland extent. The CLCD data ranged from 1985 to 2020, is the grassland extent in this study dynamic or static? Make it clear.

**Response: Thank you for your valuable comment. We used the 1990~2020 grasslands from the CLCD dataset, which has a dynamic range. We have clarified this clearly in the revised version (see L212).**

8) Line 235-239: How the adjustment was conducted?

**Response: Thank you for your question. In this study, grazing intensity is adjusted for the total amount of livestock in each county using the formula below. We have added this information in the revised version (see L240~246).**

9) Line 282: The determination coefficient ($R^2$) is 1 at county-level, indicating your model is really good. This was because you corrected the grazing intensity map (section 2.3.4). How was the model performance if there was no correcting step?

**Response: We appreciate this valuable comment. As you have noted in your above comment, the main reason for the determination coefficient of 1 is that we carried out the correction at the county scale. In the revised version, we have deleted this section and the related statement. In addition, the model performance with no correction was also showed in the second paragraph of Section 3.1 (see L278~282 in the main text).**

10) Line 379-381: Figure 7: The grazing intensity data from Meng et al. (2023) is not consistent with the original figure, especially the spatial extent. Please check it carefully.

**Response: Thanks for raising this point. We think that for grazing maps, only areas with grazing intensity greater than zero are meaningful. Therefore, for the grazing intensity data provided by Meng et al. (2023), we extracted only those areas in which the grazing intensity was greater than zero. As a result, the maps appearing in our text are not consistent with their original maps. We hope you find these explanation rise to your expectations.**

11) Section 4.2 Implications for grazing management. This section needs to be reorganized and add more findings (e.g., the hotspots of grazing) from the newly developed grazing intensity datasets rather than just talked the driving forces of spatial heterogeneity of grazing intensities.

**Response: Thanks for raising this important point. We totally agree with you. In response, we provide an in-depth discussion of grazing management based on our newly developed grazing intensity maps on the QTP in the revised version. We hope you find these revisions rise to your expectations (see L427~489).**

12) Line 423-425: A research "High-resolution livestock seasonal distribution data on the Qinghai-Tibet Plateau in 2020" already incorporated the seasonality into livestock spatial distribution mapping in the QTP.

**Response: Thanks for your comment. Yes, there have been studies that have incorporated seasonality into livestock spatial distribution. Nevertheless, we discuss here the limitations of this study because we did not distinguish between warm-season and cold-season pastures.**

13) Line 428-430: Agree with your discussion. See the second major comment.

**Response: Thanks for this comment. We agree that due to the lack of enough training samples, the accuracy of model simulations may be compromised in this study. However, as we noted earlier in response to your second major comment, the ET model yielded relatively good results in livestock spatialization (see our response to your second major comment). We think that these limitations stated in section 4.4 are exactly where future research should focus.**

**Response to RC2**

**General Comments:**

This manuscript presents a study utilizing the Extreme Trees (ET) model and detailed census data to produce annual Gridded Dataset of Grazing Intensity (GDGI) maps spanning from 1990 to 2020 in the Qinghai-Tibet Plateau (QTP) at a resolution of 100 meters. The authors compare the performance of five machine learning algorithms in delineating spatial patterns of grazing intensity in the QTP, concluding that the ET model offers the most accurate estimation. Overall, the manuscript is well-written with clearly defined objectives, and the study itself is both interesting and well-executed. However, there are some concerns regarding the lack of clarity and justification for uncertainties in data and results, as well as a few minor points that could be addressed.

**Response: Thank you very much for your overall positive words about our manuscript. In the revised version, we have improved the manuscript according to your suggestions, especially your concerns regarding the lack of clarity and justification for uncertainties in data and results. In the following sections, you will find our detail responses to each of your points. We are grateful for the time and energy you expended on our behalf.**

**Specific comments**

The main shortcomings of this study include:

1. The description of parameters for the machine learning models you trained is lacking.

**Response: Thanks for raising this important point. In the revised version, we have introduced more detail information about the five machine learning algorithms as well as the description of parameters in the appendix (see L148~182 in the supplement file for details).**

2. What is the optimal model you used in Line 235?

**Response: Thank you for your question. When introducing the methodological framework for mapping high-resolution grazing intensity, we did not know which of the five models performed best. Thus, we can not specify which model was optimal at this section. In section 3.1 (Performances of models), according to the simulation verification results, it can be seen that the ET model performs best among the five models.**

3. The authors have made efforts to address uncertainties and limitations in the discussion section. However, I find that the treatment of these aspects is somewhat superficial and lacks depth. There are uncertainties from input data,

model, and the framework. I expect when you offer the final predictions with uncertainties.

**Response: Thanks for raising this important point. We totally agree with you. In response, we provide an in-depth discussion of uncertainties and limitations in the revised version, from input data, model as well as the framework (see L490~535 in the main text). We hope you find these revisions rise to your expectations.**

4. In Table 1 and Figure 3, the R2 is too high, which may not accurately reflect real-world conditions. To ensure reproducibility and transparency, it would be beneficial to provide access to the model training and validation data, as well as the code used in the study.

**Response: We appreciate this valuable comment. In the revised version, we have provided both the training and validation data in the appendix (see Table S4 in the supplement file for details). Nevertheless, due to the confidentiality requirements of the relevant departments providing the livestock census data, we can not specify the exact name of each county. However, this does not affect the validation process of the model. Moreover, the code used in this study can be found in https://figshare.com/s/ad2bbc7117a56d4fd88d. We hope you find these revisions rise to your expectations.**

Minor issues:

1. "Random Forests" is the right name for the algorithm.

**Response: Thanks for raising this point. We have correct "Random Forest" as "Random Forests" in the revised version (see through the text).**

2. Line 443 Conclusions should be 6.

**Response: Done. Thank you (see L544).**

3. 5 Data availability could be moved after the conclusions.

**Response: According to the requirements of ESSD, the section of Data availability is usually placed before Conclusions.**

**Response to RC3**

**General Comments:**

Grazing has been an important human activity in the grasslands of the Tibetan Plateau for thousands of years. If grazing intensity is not considered as a factor, then most ecological problems cannot be solved. Therefore, this study has made an important attempt to do so, and in particular, has made a commendable effort by providing a map of grazing intensity on the plateau. However, I have several reservations about the research methodology used in the paper, which is directly related to the results of the study.

**Response: Thank you very much for recognizing the value of our paper and for your overall positive words about it. We are delighted to hear that our manuscript received such feedback. In the revised version, we have improved the manuscript according to your comments and suggestions, especially your concerns regarding the research methodology. In the following sections, you will find our detail responses to each of your points and comments. We are grateful for the time and energy you expended on our behalf.**

**Specific comments**

First, discrepancy in spatial resolution across nodel phases. This approach translates county-level analysis in model training and validation to grid-level (100 m resolution) prediction. One problem with this approach is the applicability of county-level training models to grid-level predictions without rigorous validation at the same finer resolution. Whilst the challenge of obtaining grid-level validation data is recognised, it is essential to ensure the reliability of model predictions at this scale. I recommend a detailed revision of the methodology to address or justify this discrepancy. This is where the study comes to life.

**Response: Thanks for raising this important point. We totally agree with your comment and have incorporated pixel-scale validation of the GDGI dataset to ensure the model's predictive accuracy at this granularity (see L 289-298 in the main text). The pixel-scale validation dataset comprises a total of 112 records across 68 locations, which were derived from literatures, questionnaires, and field investigations. Furthermore, comprehensive details about these ground-truth validation data are provided in the appendix (see Figure S3 and Table S3 in the supplement file for further information).**

Second, issues with averaging ecological factors at the county level. The study's approach of averaging environmental and other factors at the county level to infer grazing intensity raises questions about its validity and accuracy. Given the variability in county size and environmental conditions, this approach may oversimplify the

complex dynamics of grazing intensity. A more nuanced approach, which might include feature selection and evaluation, might better capture the different influences on grazing intensity in different counties. In addition, the hypothesised correlation between average factors and grazing intensity deserves further research to confirm its validity.

**Response: This is another important point. We fully acknowledge the legitimacy of the concern that averaging environmental factors might oversimplify the intricate dynamics between grazing intensity and lead to a certain degree of estimation inaccuracies. In our research, the rationale behind employing county-scale averages of environmental factors is to align with county-scale livestock census data. Although theoretically, it would be ideal to utilize sub-county scale livestock census data to cover the entire Tibetan Plateau, practical limitations currently render this approach nearly unfeasible. Consequently, grazing maps have widely been modeled through the averaging of environmental factors, predicated on the hypothesis that a consistent causal relationship between livestock intensity and environmental factors persists across various scales (Robinson et al., 2014; Nicolas et al., 2016; Li et al., 2021; Meng et al., 2023). Nonetheless, as far as the actual situation is concerned, the influence of environmental factors on livestock distribution varies across different scales, a fact we have addressed as a potential limitation in the revised manuscript (refer to L505~510 in the main text for details).**

[Figure]

Figure R5 Importance of environmental factors influencing the spatial distribution of grazing intensity.

**Indeed, we had previously identified this limitation. To mitigate it as effectively as possible, we assessed the relative significance of each environmental factor in predicting livestock distribution at the county scale before initiating the**

estimation process (see Figure R5 in the response attachment). The analysis revealed that the seven environmental factors collectively account for 91.94% of the variance in grazing intensity distribution, with all factors achieving statistical significance ($p<0.05$). This demonstrates a specific correlation between the selected environmental factors and livestock distribution, underscoring their potential predictive power regarding grazing intensity.

Further validation at the pixel scale corroborates our approach. The validation results demonstrate that the GDGI dataset maintains high accuracy at this finer resolution (see L 289-298 in the main text), further validating the use of county-scale averaged environmental factors in predicting livestock distribution.

Third, exclusion of vegetation productivity factors: The omission of productivity indicators such as grass biomass or grass quality from the model is a serious flaw. These factors are intrinsically linked to grazing intensity and could greatly improve the explanatory power of the model. Without quality control of the grassland, all grazing is not possible. This may be the determining factor in whether the paper is published. It is therefore recommended that the relationship between grassland quality and grazing intensity be assessed to gain a fuller understanding of the determinants affecting grazing intensity and patterns in alpine grasslands.

Response: Thanks for raising this point. We apologize for any confusion. Indeed, our study has taken into account factors of vegetation productivity, opting for the Human-activity-induced Net Primary Productivity (HNPP) indicator over the traditional Net Primary Productivity (NPP). Numerous studies have validated NPP as a crucial metric for assessing grassland biomass and quality (Scurlock et al., 2002; Cheng et al., 2023; Adam et al., 2024). Our preference for HNPP over NPP is twofold.

Initially, it is critical to acknowledge that grassland biomass or quality merely indicates potential rather than actual grazing intensity. In other words, while grassland NPP may theoretically correlate with potential grazing capacity, it does not have a direct relationship with actual grazing activities. For instance, grasslands within nature reserves on the Tibetan Plateau often exhibit higher NPP compared to those in non-protected areas, yet due to strict restrictions for grazing activities, leading to lower grazing intensities (Zhu et al., 2023).

Furthermore, on the Tibetan Plateau, grazing activities represent the predominant form of grassland utilization. As mentioned above, HNPP quantifies the actual primary productivity harvested through human activities, thus reflecting the real extent of biomass utilization by humans. In addition, evidence suggests that HNPP bears a closer association with grazing activities than does NPP (Huang et al., 2022; Zhou et al., 2024).

For these reasons, we employed HNPP as a proxy indicator for grass biomass and quality in this study. Our model, integrating the HNPP indicator, exhibited commendable performance, elucidating 95.4% of the grazing intensity with

**notable robustness (Figure 3 in the main text). In response to your feedback, we have further clarified our rationale for selecting HNPP over NPP in the revised manuscript (see L122-130 and L390-392 for detailed explanation). We hope you find these revisions rise to your expectations.**

Other issue. From Fig. 5, around 1996 to 2000, government statistics show a dramatic reduction in grazing intensity, which may be caused by a change in statistical standards or policy, not necessarily a real reduction in grazing intensity. This needs to be explained in detail, otherwise there would be such a sharp change and the predictive model should be able to catch it.

**Response: We appreciate this valuable comment. Yes, as shown in Figure 5 (Figure 7 in the revised version), the average grazing intensity on the QTP experienced a pronounced decrease in around the 1996-2000 period. Our investigation confirms that this significant reduction is not attributed to changes in statistical standards but rather to a substantial decrease in livestock numbers. Specifically, government livestock census data indicate a reduction of approximately 1,556,000 sheep units in livestock populations across the Tibetan Plateau during this period. The observed sharp decline in around 1996-2000 also aligns with other studies (Ye et al., 2020; Sun et al., 2022; Meng et al., 2023). Encouragingly, our GDGI maps have accurately captured this sharp change (refer to Figure 7d in the revised version). In light of your feedback, we have provided a thorough analysis of this decline in the discussion section (see L417-426 in the main text for details).**

**References**

Adam, T.N., Steven R.A. and Philip H. (2024). Comparing the Predictive Capacity of Allometric Models in Estimating Grass Biomass in a Desert Grassland. *Rangeland Ecology & Management*, 93, 72-76. https://doi.org/10.1016/j.rama.2024.01.004.

Cheng, Z., Zhao, J., Ding, L., Shi, Z., Gao, P., and Wu, L. (2023). The functioning of alpine grassland ecosystems: Climate outweighs plant species richness. *Journal of Ecology*, 111, 2496-2506. https://doi.org/10.1111/1365-2745.14202.

Huang, X., Yang, Y., Chen, C., Zhao, H., Yao, B., Ma, Z., Ma, L., and Zhou, H. (2022). Quantifying and Mapping Human Appropriation of Net Primary Productivity in Qinghai Grasslands in China. *Agriculture*, *12*(4), 483. https://doi.org/10.3390/agriculture12040483

Li, X., Hou, J., and Huang, C. (2021). High-Resolution Gridded Livestock Projection for Western China Based on Machine Learning. *Remote Sensing*, *13*(24), 5038. https://doi.org/10.3390/rs13245038

Meng, N., Wang, L., Qi, W., Dai, X., Li, Z., Yang, Y., Li, R., Ma, J., and Zheng, H. (2023). A high-resolution gridded grazing dataset of grassland ecosystem on the Qinghai-Tibet Plateau in 1982-2015. *Scientific Data*, *10*(1), 68. https://doi.org/10.1038/s41597-023-01970-1

Nicolas, G., Robinson, T. P., Wint, G. R., Conchedda, G., Cinardi, G., and Gilbert, M. (2016). Using Random Forest to Improve the Downscaling of Global Livestock Census Data. *PLoS One*, *11*(3), e0150424. https://doi.org/10.1371/journal.pone.0150424

Robinson, T. P., Wint, G. R., Conchedda, G., Van Boeckel, T. P., Ercoli, V., Palamara, E., Cinardi, G., D'Aietti, L., Hay, S. I., and Gilbert, M. (2014). Mapping the global distribution of livestock. *PLoS One*, *9*(5), e96084. https://doi.org/10.1371/journal.pone.0096084

Scurlock, J.M.O., Johnson, K. and Olson, R.J. (2002). Estimating net primary productivity from grassland biomass dynamics measurements. *Global Change Biology*, 8: 736-753. https://doi.org/10.1046/j.1365-2486.2002.00512.x

Sun, Y., Liu, S., Liu, Y., Dong Y., Li, M., An, Y., and Shi, F. (2022). Grazing intensity and human activity intensity data sets on the Qinghai-Tibetan Plateau during 1990–2015. *Geoscience Data Journal*, J. 9, 140–153. https://doi.org/10.1002/gdj3.127

Ye, T., Liu, W., Mu, Q., Zong, S., Li, Y., and Shi, P. (2020). Quantifying livestock vulnerability to snow disasters in the Tibetan Plateau: Comparing different modeling techniques for prediction. *International Journal of Disaster Risk Reduction*, 48, 101578. https://doi.org/10.1016/j.ijdrr.2020.101578

Zhou, W., Wang, T., Xiao, J., Wang, K., Yu, W., Du, Z., Huang, L., and Yue, T. (2024). Grassland productivity increase was dominated by climate in Qinghai-Tibet Plateau from 1982 to 2020. *Journal of Cleaner Production*, *434*. https://doi.org/10.1016/j.jclepro.2023.140144

Zhu, Q., Chen, H., Peng, C., Liu, J., Piao, S., He, J.-S., Wang, S., Zhao, X., Zhang, J., Fang, X., Jin, J., Yang, Q.-E., Ren, L., and Wang, Y. (2023). An early warning signal for grassland degradation on the Qinghai-Tibetan Plateau. *Nature Communications*, *14*(1), 6406. https://doi.org/10.1038/s41467-023-42099-4

---

## Author Response (AR2)

**Response to Reviewer 1**

**General Comments*:**

This manuscript developed an annual grazing intensity dataset at 100-m spatial resolution for the Qinghai-Tibet Plateau (QTP) by integrating machine learning algorithms, census data, and multiple environmental and socio-economic data. Through comparisons with previous datasets, census data, and field observations, the GDGI data proved to be a good grazing intensity dataset with fine resolution. The data can be applied to quantify the impacts of grazing management on the ecosystem in the QTP. The authors have done good work in response to the comments from the reviewers and did lot of work on input data preparation, field-level validation, and discussion sections. However, some concerns about data uncertainties and a few minor points that could be addressed.

**Response: Thank you very much for your overall positive words about our revised paper. We are delighted to hear that our manuscript received such feedback. We have improved the manuscript according to your suggestions, especially your concerns about data uncertainties. In the following sections, you will find our detail responses to each of your points and suggestions. We are grateful for the time and energy you expended on our behalf.**

**Major comments:**

According to the methods, the ET model predicted mean grazing intensity at the county level was corrected in the final step, but it can't solve the problem of spatial uncertainty. Thus, the authors should also generate the uncertainty maps by using the ET model and provide to the data users.

**Response: We appreciate your valuable suggestion and concur that the inclusion of uncertainty maps is essential. In response to your feedback, we have employed the ET model in conjunction with Monte Carlo simulation techniques to generate the uncertainty maps (refer to sections 2.5 and 3.2 in the revised version).**

**In addition, we have also provided the original simulation results from the ET model, along with the corresponding error maps (refer to Figure 4c).**

Additionally, I found that the GDGI data (https://figshare.com/s/ad2bbc7117a56d4fd88d, https://zenodo.org/records/10851120) was not publicly available, and I couldn't download the data.

**Response: We apologize for this inconvenience due to the expired link. Please check the updated link below. https://zenodo.org/records/13141090?preview=1&token=eyJhbGciOiJIUzUxMiJ**

*Some minor comments:*

Line 280: Is it '0123'? Please check it.

**Response: Thanks for raising this point. We have corrected "0123" as "0.123" (see line 308 in the revised version).**

Figure 5: Why do about half of the field observations in the Alpine Meadow ecosystem have the same grazing intensity (about 4 Su/ha)? Please add some short explanations.

**Response: We are grateful for your insightful comment. In direct response to your concerns, we have conducted a meticulous review of all field observations. Our verification confirms that the grazing intensity in the alpine meadows predominantly falls within the range of $4 \pm 0.5$ SU/ha on the Qinghai-Tibet Plateau. This range is supported by findings from other peer-reviewed scholarly articles that are grounded in extensive, long-term observational studies (Cao et al., 2004; Li et al., 2018a; Wang et al., 2020; Zhuang et al., 2019). Consequently, we are confident that this data accurately represents the actual grazing conditions within the alpine meadows of the Plateau (refer to Table S3 for details).**

Figure 7e should include a legend of linear trend value.

**Response: We appreciate your observation. In Figure 7e, rather than presenting the linear trend values in isolation, we have chosen to depict them in conjunction with their statistical significance. Specifically, a linear trend with a value greater than zero and a p-value less than 0.01 is illustrated in dark purple, whereas a trend with a value less than zero and a p-value less than 0.01 is depicted in dark green (refer to Figure 8e in the revised version).**

**Response to Reviewer 2**

*General Comments:*

Enriching the grid data on grazing intensity on the Qinghai-Tibet Plateau is significant for improving environmental impact assessments. While the paper has seen substantial improvements after several revisions, the data still shows considerable uncertainty.

**Response: We are deeply grateful for your acknowledgment of the significance of our work. In the updated manuscript, we have carefully incorporated your valuable feedback. In particular, to address your concerns about data uncertainty, we have added new sections (please refer to sections 2.5 and 3.2 for details in the revised version) that utilize an ET model in conjunction with a Monte Carlo simulation to evaluate the uncertainty within the annual grazing intensity maps. In the subsequent sections, we provide a detailed response to each of your perceptive comments. It is our earnest hope that these revisions meet your expectations.**

*Major comments:*

Firstly, in Section 3.1, the performance metrics of the validation set are excellent. However, in Section 3.2, the validation results of the calibrated grazing data are worse. Why is this? Is it related to the choice of the model's test set, or are there other factors? Moreover, the data for township validation points is very sparse. Is it reasonable to increase samples through a temporal dimension?

**Response: Thanks for raising this important point. We appreciate your insight regarding the potential impact of test set selection on model performance. To address this concern, we conducted additional verification. Specifically, we employed Python's train_test_split function from the sklearn. Model_selection library to randomly allocate 30% of the data as the test set. To assess the influence of test set selection on model performance, we varied the random_state parameter from 1 to 20 and executed the model 20 times. Consistently, the range of R² values was between 0.92 and 0.97, suggesting that the choice of test set does not substantially affect the model's performance in this study.**

**Regarding the variance in model performance which you mentioned, we attribute the primary cause to the scale effect. In Section 3.1, we demonstrate the model's performance on a county-scale test set comprising 30% of the total, with 1499 samples, yielding an R² value of 0.955. In Section 3.3 of the revised version**

(formerly Section 3.2), the model's performance on the township scale, involving 18 townships, resulted in an R² value of 0.867; on the pixel scale, with 112 points, the R² value was 0.804. This indicates a decline in model performance as the scale becomes more refined. The primary reason for this may be the resolution of the input data. Since the census data in model training is county-level, the input environmental variables also correspond to county-level, which to some extent, smooths spatial details and thus limits the model's expressiveness at finer scales.

We acknowledge the scarcity of township-scale validation data in our study. As you know, collecting livestock census at the township level in the Tibetan Plateau is indeed challenging. We have highlighted this limitation in the discussion section of the revised version (see lines 577~581). To address the issue of insufficient validation data from the same year, we independently verified the data across different years on a temporal scale. In fact, in cases of limited validation data, scholars frequently increase samples through a temporal dimension. For instance, Meng et al. (2023) validated their grazing data using livestock data from 2001 to 2021(Meng et al., 2023), and Venter et al. (2016) used satellite imagery data from 1999 to 2015 to validate their human footprint data. We hope you find these revisions rise to your expectations (Venter et al., 2016).

Secondly, in Figure 7, there is a sharp decline in the data from 1990 to 2000. What is the cause? Given this significant decline, does it contradict the common understanding that China implemented large-scale ecological projects in the new century? I suggest starting the data development from the year 2000.

Response: This is another good point. Yes, Figure 7 (Figure 8 in the revised version) illustrates a sharp decline in grazing intensity on the Qinghai-Tibet Plateau from 1997 to 2001. This downward trend is corroborated by official livestock census from governments, which indicate a reduction of 1,889,400 livestock during this period. Furthermore, this trend also aligns with findings from other researches. For instance, Sun et al. (2021) noted a decrease in grazing intensity of 43 sheep units per hectare from 1995 to 2000. The precipitous drop in grazing intensity between 1997 and 2001 is likely predominantly associated with natural disasters. Take Naqu alone as an example, Ye et al. (2020) demonstrated that the severe snow disaster of 1997–1998 has led to the loss of more than 820,000 livestock (Ye et al., 2020).

Figure 7 (Figure 8 in the revised version) also delineates a significant increase in grazing intensity on the Qinghai-Tibet Plateau from 2000 to 2010. This

observation is congruent with the results reported by Sun et al. (2021) and Li et al. (2021) (Li et al., 2021; Sun et al., 2021). This increase in grazing intensity is likely a rebound effect following the natural disasters. Additionally, Figure 7 (Figure 8 in the revised version) indicates a consistent decline in grazing intensity across the plateau post-2010, which is largely attributed to the implementation of extensive ecological projects. To address any potential reader misconceptions about this phenomenon, it has been thoroughly discussed in the revised draft's discussion section (refer to lines 461-473).

Thirdly, while the authors emphasize the high spatial resolution of the developed data, only the elevation, slope, and population data among the auxiliary data have a comparable resolution. Furthermore, the evaluation of the population data shows that its accuracy on the plateau is not high. In this context, higher spatial resolution may lead to greater uncertainty. Do the authors need such a high resolution?

Response: We appreciate your raising this crucial point. We are well aware of the formidable challenge in generating a 100-meter resolution map of grazing intensity across the Tibetan Plateau. Our pursuit is underpinned by three key motivations. Firstly, we firmly believe that high-resolution grazing maps will substantially improve grassland management and guide pertinent decision-making on the plateau. Secondly, the successful publication of several global-scale 100-meter resolution maps by other scholars has underscored the feasibility and reliability of this resolution in practical applications. For example, studies on population distribution, forest management, and land cover have adeptly utilized this resolution, as evidenced by the works of Lloyd et al. (2017, Lesiv et al. (2022), and Masiliūnas et al. (2021) (Lesiv et al., 2022; Lloyd et al., 2017; Masiliūnas et al., 2021). Lastly, the key input data for our grazing dataset, though generated by ourselves, have demonstrated their efficacy through stringent accuracy validation (refer to lines 49-112 in the supplementary file). These results reinforce our confidence in producing a 100-meter resolution grazing map that promises to be both precise and valuable.

Regarding the accuracy of the population distribution data for the Qinghai-Tibet Plateau, which you highlighted, we selected the 100-meter resolution data from Worldpop, primarily informed by the research of Li et al. (2020). Their comparative analysis of WorldPop (100m), GPW (1000m), and the Chinese datasets CnPop1, CnPop2, and CnPop3 (all at 1000m resolution, except CnPop3 at 100m) within the Qinghai-Tibet Plateau revealed that the WorldPop data at 100m resolution offers the highest degree of accuracy, with an $R^2$ value in

relation to county-level census data that can even approach 0.90(Li et al., 2020). That is to say, WorldPop data has a good performance in the Tibetan Plateau, which is also the reason why we use it as the data source of population distribution.

Regarding the constraints, why are areas with a population density greater than 50 people/km² designated as non-grazing areas? Grazing areas often overlap with rural residential areas, and livestock often graze around these settlements in the mornings and evenings, leading to higher grazing intensity. Additionally, in the Qinghai-Tibet Plateau, there are many uninhabited areas where animals often live. Also, how is slope related to grazing areas? What is the basis for the 40° threshold?

Response: We appreciate this valuable comment. In this study, we define areas with a population density exceeding 50 people/km² as non-grazing areas, primarily drawing on the research conducted by Li et al. (2018). In their study, areas with a population density above 50 people/km² were categorized as urban built-up areas on the Qinghai-Tibet Plateau (Li et al., 2018b). We consider it logical to extend this definition to classify urban areas as non-grazing zones. In our research, all regions falling below this population density were designated as potential grazing areas, including the uninhabited areas you mentioned, ensuring that the distribution of rural settlements is fully accounted for and the continuity of grazing practices is maintained.

The rationale for designating 40% of the area as non-grazing is largely informed by the findings of Robinson et al. (2014). In their work on the Gridded Livestock of the World (GLW2), they suggested that areas with slopes exceeding 40% were unsuitable for grazing activities due to topographic constraints and issues of accessibility (Robinson et al., 2014). We have adopted this criterion to ensure that our model incorporates the impact of topographic factors on the feasibility of grazing activities in our assessments of grazing suitability. We hope you find these explanations rise to your expectations.

In Figure 4, the readability of the image details is insufficient.

Response: We appreciate your observation. In response, we have revised Figure 4 to enhance its clarity and improve readability for readers (see Figure 4).

Finally, as a data-centric paper, I recommend including the selection and description of auxiliary variables in the main text.

Response: We appreciate this valuable suggestion. In the revised version, we have added the selection and description of auxiliary variables in the main text (see section 2.2).

**Response to Reviewer 3**

*General Comments:*

This study provides a high-resolution grazing intensity dataset on the Qinghai-Tibet Plateau using machine learning algorithms based on county-level data. The dataset could be a potential interest to other researchers who focus on the Tibet Plateau. Authors also compare and validate their datasets with other comparable datasets as well as their own monitoring data. The dataset is also accessible. I still found more clarifications/corrections needed before it can be considered for publication.

**Response: Thank you for these overall positive words. We have thoroughly addressed all of your comments and suggestions in the revised version. Please see our detail responses to the individual comments in the following sections. We are grateful for the time and energy you expended on our behalf.**

*Major concerns:*

1. Clarify the reason for 100 m. Though 100m spatial resolution is appealing, the authors didn't clarify why they chose this spatial resolution for the final dataset. Actually, if I understood correctly, the authors trained and validated the model at the county level and then applied the model to 100m spatial resolution. (a) What is the reason for this? Do the authors have a reliable input at the 100m spatial resolution?

**Response: Thanks for raising this important point. We recognize the immense challenge of creating a 100-meter resolution map depicting grazing intensity across the Tibetan Plateau. Our endeavor is driven by three principal motivations. Firstly, we are convinced that high-resolution grazing maps will significantly enhance grassland management and inform related decision-making processes on the plateau. Secondly, the successful publication of several global-scale 100-meter resolution maps by other researchers have demonstrated the feasibility and reliability of this resolution in practical applications. For instance, studies on population distribution (Lloyd et al., 2017), forest management (Lesiv et al., 2022), and land cover (Masiliūnas et al., 2021) have all utilized this scale effectively (Lesiv et al., 2022; Lloyd et al., 2017; Masiliūnas et al., 2021). Lastly, the key input data for our grazing dataset, although generated by ourselves, have proven their efficacy through rigorous accuracy validation (see lines 49-112 in the supplement for details). These results bolster our confidence in producing a 100-meter resolution grazing map that will be both accurate and useful.**

(b) Actually, I also think that an overview table of all the input data is helpful, which

shows the temporal and spatial resolution of each input data (for county-level model training and 100m application respectively).

**Response: We appreciate this valuable comment. In the revised version, we have provided an overview table of each input data (see Table 1 and Table 2 in the revised version).**

(c) I also find it would be helpful to add a data use note on how users can use the data at different spatial resolutions and how to avoid potential misuse.

**Response: Thank you very much for your valuable suggestion. In the revised version, we have provided the data use notes for users, and uploaded it together with the GDGI dataset (see notes in the meta-data file).**

2. Lines 357-390 and figure 8, the discussion is not convincing. Yes, the authors' dataset shows a better alignment with observations. This could be the result of a better algorithm or selective factors, but also could simply be due to the grazing map correction based on county-level data. For a fair comparison, the authors should also correct other datasets using the county-level data and compare again.

**Response: We greatly appreciate this insightful comment. Our decision not to compare all grazing datasets after the county-level livestock census correction stems from three main considerations. Firstly, it is an objective reality that discrepancies between different grazing map products are inevitable due to differences in the methodologies used, environmental factors selected, and livestock census data employed. Because all the four grazing data products compared in this paper have been refined using control method, further corrections would alter the intrinsic values of these maps, potentially compromising their original integrity. Thus, preserving the original data values is crucial to maintaining the authenticity and scientific merit of each dataset.**

**Secondly, maintaining the original data values can ensure that comparative analyses accurately reflect the distinctions between various grazing intensity products, enhancing the transparency and reproducibility of these datasets. This approach allows readers and peers to directly assess the comparative strengths and limitations of each product through unaltered comparative results.**

**Thirdly, extensive literature review indicates that direct comparison of different data products is a normal practice. For instance, Meng et al. (2023) directly compared their grazing intensity maps with ALCC and GLW datasets. Similarly, Li et al. (2021) made a direct comparison of their grazing maps to**

GLW data(Li et al., 2021). This method is also prevalent in the analysis of other data types, as evidenced by Mu et al. (2022) who compared their human footprint data directly with three published datasets, and Li et al. (2020) who directly compared their 2010 population distribution data with five other published datasets (Li et al., 2020; Mu et al., 2022).

Therefore, in light of these precedents, we deem it more appropriate to directly compare the different grazing datasets without altering their original map product values.

To bolster the fairness and rigor of our study, in the revised version, we have standardized the verification of the GDGI products and other grazing intensity maps against livestock census data at the township level. Specifically, we have chosen livestock census data from ten townships in 2015 as the validation baseline (refer to lines 393-397 and Figure 9c in the revised version). The selection of data from 2015 is due to the availability of the GLW data product for that year only. The township-level validation results underscore that the GDGI products not only excel in spatial and temporal resolution but also exhibit the lowest error rates (see lines 396-397 in the revised version), suggesting that their superior performance attributable to a more refined algorithm and the selection of more pertinent environmental factors.

Nevertheless, we acknowledge that the livestock census from the ten townships may be limited. This limitation is addressed and discussed in the revised paper (refer to lines 578-581). We hope that our responses and revisions adequately address your concerns.

*Minor concerns:*

1. Even though the authors described the meta-data in the manuscript, I expected that the meta-data should also be included in the dataset, including the unit, coverage, and projected coordinate system.

Response: Thank you for this great suggestion. In the revised version, we have provided a meta-data file in the GDGI dataset, which contains all specific information you mentioned (see the meta data file for details).

2. Line 216, why chose 50 as a threshold?

Response: Thanks for raising this point. The determination of this threshold is based on the research results of Li et al. (2018), who regard the population

density greater than 50 people/km$^2$ as the urban building area (Li et Al., 2018). We consider it logical to extend this definition to classify urban areas as non-grazing zones.

3. Line 243, can the authors provide L_CCensus/L_Cgrid for each county and year in the supplement or their dataset?

Response: We appreciate your raising this significant point. However, due to confidentiality agreements we've established with the pertinent government departments when procuring the livestock census data, we are constrained from disclosing the original livestock data. As an alternative, in compliance with your suggestion, the revised manuscript now includes the converted grazing intensity grid data spanning the years 1990 to 2020 (see table 4S in the supplementary file).

4. How did the authors tune the parameters of each machine learning algorithm?

Response: Thanks for raising this point. The tuning of the optimal parameters of each machine learning algorithm were done in python using Randomized Search CV functions.

5. Line 228, the authors used 70% of the samples for training and the remaining for testing for the algorithm selection. After selecting the Extra Trees regression as the best algorithm, did you train the model with all the samples (100%) again before applying it to 100m spatial resolution?

Response: This is indeed another critical point. Following the selection of the Extra Trees regression model as the optimal algorithm, we did not retrain the model using all samples. This decision was based on the testing results, which showed an R-squared (R²) value as high as 0.955, indicating that the ET model performs exceptionally well. However, we fully concur with your view that the reliability of the ET model should be validated before its application at a 100-meter resolution. To this end, in the revised manuscript, we have conducted 100 random simulations using the Monte Carlo method at the 100-meter spatial scale to further ascertain the robustness of the ET model at this resolution. The Mean Relative Error (MRE) and Standard Deviation (STD) of the simulation outcomes are also found to be satisfactory, thereby further substantiating the model's reliability at the 100-meter spatial resolution (refer to lines 314-326). We hope that these revisions will meet your expectations.

6. The authors may consider including the sample data from the supplement to their dataset to increase the accessibility.

**Response: Thank you for your good suggestion. In the revised version, we have included the sample data to the GDGI database.**

7. Supplement, line 28, disaggregate?

**Response: Done. Thank you (see line 28 in the supplement).**

**References**

Cao, G. M., Tang, Y. H., Mo, W. H., Wang, Y. S., Li, Y. N., and Zhao, X. Q., 2004. Grazing intensity alters soil respiration in an alpine meadow on the Tibetan plateau. Soil Biol Biochem. 36, 237-243.

Lesiv, M., Schepaschenko, D., Buchhorn, M., See, L., Dürauer, M., Georgieva, I., Jung, M., Hofhansl, F., Schulze, K., and Bilous, A., 2022. Global forest management data for 2015 at a 100 m resolution. Sci Data. 9, 199.

Li, G., Zhang, Z., Shi, L. L., Zhou, Y., Yang, M., Cao, J. X., Wu, S. H., and Lei, G. C., 2018a. Effects of Different Grazing Intensities on Soil C, N, and P in an Alpine Meadow on the Qinghai—Tibetan Plateau, China. Int J Env Res Pub He. 15, 2584.

Li, L., Zhang, Y., Liu, L., Wang, Z., Zhang, H., Li, S., and Ding, M., 2020. Mapping Changing Population Distribution on the Qinghai–Tibet Plateau since 2000 with Multi-Temporal Remote Sensing and Point-of-Interest Data. Remote Sens., 4059.

Li, S., Wu, J., Gong, J., and Li, S., 2018b. Human footprint in Tibet: Assessing the spatial layout and effectiveness of nature reserves. Sci Total Environ 621, 18-29.

Li, X. H., Hou, J. L., and Huang, C. L., 2021. High-Resolution Gridded Livestock Projection for Western China Based on Machine Learning. Remote Sens. 13, 5038.

Lloyd, C. T., Sorichetta, A., and Tatem, A. J., 2017. High resolution global gridded data for use in population studies. Sci Data. 4, 1-17.

Masiliūnas, D., Tsendbazar, N.-E., Herold, M., Lesiv, M., Buchhorn, M., and Verbesselt, J., 2021. Global land characterisation using land cover fractions at 100 m resolution. Remote Sens Environ. 259, 112409.

Meng, N., Wang, L. J., Qi, W. C., Dai, X. H., Li, Z. Z., Yang, Y. Z., Li, R. N., Ma, J. F., and Zheng, H., 2023. A high-resolution gridded grazing dataset of grassland ecosystem on the Qinghai-Tibet Plateau in 1982-2015. Sci Data. 10, 68.

Mu, H., Li, X., Wen, Y., Huang, J., Du, P., Su, W., Miao, S., and Geng, M., 2022. A global record of annual terrestrial Human Footprint dataset from 2000 to 2018. Sci Data 9, 176.

Robinson, T. P., Wint, G. R., Conchedda, G., Van Boeckel, T. P., Ercoli, V., Palamara, E., Cinardi, G., D'Aietti, L., Hay, S. I., and Gilbert, M., 2014. Mapping the global distribution of livestock. Plos One. 9, e96084.

Sun, Y. X., Liu, S. L., Liu, Y. X., Dong, Y. H., Li, M. Q., An, Y., and Shi, F. N., 2021. Grazing intensity and human activity intensity data sets on the Qinghai-Tibetan Plateau during 1990–2015. Geosci

Data J. 9, 140-153.

Venter, O., Sanderson, E. W., Magrach, A., Allan, J. R., Beher, J., Jones, K. R., Possingham, H. P., Laurance, W. F., Wood, P., Fekete, B. M., Levy, M. A., and Watson, J. E., 2016. Global terrestrial Human Footprint maps for 1993 and 2009. Sci Data 3, 160067.

Wang, J. L., Liu, Y. Z., Cao, W. X., Li, W., Wang, X. J., Zhang, D. G., Shi, S. L., Pan, D. F., and Liu, W. L., 2020. Effects of grazing exclusion on soil respiration components in an alpine meadow on the north-eastern Qinghai-Tibet Plateau. Catena. 194, 104750.

Ye, T., Liu, W. H., Mu, Q. Y., Zong, S., Li, Y. J., and Shi, P. J., 2020. Quantifying livestock vulnerability to snow disasters in the Tibetan Plateau: Comparing different modeling techniques for prediction. International Journal of Disaster Risk Reduction 48.

Zhuang, M. H., Gongbuzeren, Zhang, J., and Li, W. J., 2019. Community-based seasonal movement grazing maintains lower greenhouse gas emission intensity on Qinghai-Tibet Plateau of China. Land Use Policy. 85, 155-160.

---

## Author Response (AR3)

**Response to Reviewer 1**

All my major concerns have been well addressed in the revised manuscript. I recommend accepting this manuscript for publication in Earth System Science Data.

I can access the grazing intensity dataset in the Zenodo platform using the link provided by the authors, but the dataset is not publicly available using the following link (https://zenodo.org/records/13141090). Additionally, the grazing intensity uncertainty dataset could also be uploaded.

**Response: We appreciate your comments and suggestions. The grazing intensity dataset, which can be accessed via https://zenodo.org/records/13141090, is currently under embargo. It will be made publicly available upon acceptance of this manuscript. Furthermore, we have uploaded the uncertainty data for grazing intensity from 1990 to 2020 to the grazing intensity database, which can now be accessed through the following link: https://zenodo.org/records/13701486?token=eyJhbGciOiJIUzUxMiJ9.eyJpZCI6IjA4YzVkM2FkLTA2NzktNDczYi05ZDA4LTk3ZGNjYmViYjRjZSIsImRhdGEiOnt9LCJyYW5kb20iOiJlYTIwZTg5ODIxM2M0M2E1N2UzNzQ0ZmMzMGNiNzFiMSJ9.Oqcf7bqs_Yd_u0PEBQw2e1_w-JEpP-P00qP7yRjoVb9mUof7ATdeBaXl2cIw6Tqw71QSEhDH5yrkfe1fyjK7mw**

**Response to Reviewer 2**

The authors have responded to the feedback, but there is a lack of substantial revisions. In the next round of revisions, it is recommended to address the data uncertainties, particularly for the data prior to the year 2000, in the abstract or conclusion section. Additionally, the results presented in Table 3 and Figure 3 are log-transformed, which could potentially mislead readers.

**Response: We are grateful for the time and energy you expended on our behalf. Regarding the issue of data uncertainty you raised, we have provided additional statements in the conclusion section of the revised manuscript (refer to lines 598-601). However, we do not concur that the data uncertainty is higher prior to the year 2000. In fact, as depicted in Figure 5a, the results from the Monte Carlo simulations indicate that the MRE for the period 1990-2000 ranges between 6.84% and 7.62%, whereas for the period 2001-2020, the MRE ranges between 7.33% and 9.08%. Furthermore, as we stated in our response to the previous revision, the conclusion that grazing intensity on the Qinghai-Tibet Plateau decreased between 1990 and 2000 is consistent with the findings of other scholars.**

**Additionally, Table 3 already contains the actual values, with the exponentials of the log-transformed numbers. Figure 3 does display the log-transformed results, and it is well annotated on the axes. In the revised version, to clarify any potential confusion, we have provided explanations in the text (refer to lines 295-296 and lines 298-299). We hope you find these revisions rise to your expectations.**

**Response to Reviewer 3**

First, I cannot access the latest dataset. So, I cannot conclude on the data updates. Please make sure to update the dataset link for reviewers after each update and test if the link works. Also, please attach the link to the responses as the link in the manuscript does not work.

**Response: We apologize for this inconvenience due to the expired link. Please check the updated link below.**

**https://zenodo.org/records/13701486?token=eyJhbGciOiJIUzUxMiJ9.eyJpZCI6IjA4YzVkM2FkLTA2NzktNDczYi05ZDA4LTk3ZGNjYmViYjRjZSIsImRhdGEiOnt9LCJyYW5kb20iOiJlYTIwZTg5ODIxM2M0M2E1N2UzNzQ0ZmMzMGNiNzFiMSJ9.Oqcf7bqs_Yd_u0PEBQw2e1_w-JEpP-P00qP7yRjoVb9mUof7ATdeBaXl2cIw6Tqw71QSEhDH5yrkfe1fyjK7mw**

Regarding my major comment #1 (reviewer 3 in your responses): The responses are still unconvincing. The authors trained and validated the model at the county level and then applied it to 100m resolution. Will there be some problem in this resolution transition? Did the authors assume the impacting factors at the county level are the same as the 100m level?

**Response: We appreciate your insightful comments and the feedback provided. We apologize for any lack of clarity in our previous responses. Indeed, in this study, we have adopted the assumption made by the Food and Agriculture Organization (FAO) when creating global livestock grid maps—that the relationship between grazing intensity and environmental factors is similar across both administrative and pixel scales (refer to lines 214-215). This assumption underpins the creation of the majority of current grazing intensity maps (Robinson et al., 2014; Li et al., 2021; Liu, 2021; Zhan et al., 2023). We acknowledge that this approach to scale conversion may introduce certain limitations, as it inevitably smooths spatial details, thereby constraining the model's expression at the pixel scale. In recognition of this limitation, we have provided a detailed discussion in the discussion section (refer to lines 552-556). We hope you find these explanations rise to your expectations.**

Regarding my major comment #2: What I meant is that the authors should weaken their statement on using a better algorithm and better factor selection as the comparisons are not the direct evidence to support their argument.

**Response: We appreciate your important observation and concur with your suggestion. In the revised manuscript, we have moderated our assertion regarding the employment of a superior algorithm and the selection of more effective factors. Furthermore, we have expanded our discussion to consider that the choice of algorithm and factors might indeed be contributing factors to the discrepancies observed between the maps (refer to lines 421-427).**

Regarding my minor comment #5: Would the easiest way to address this issue is to train the model with 100% samples rather than run the model 100 times to see the variance?

**Response: We appreciate your recurring inquiry on this matter. The decision not to utilize 100% of the samples for training the ET model was deliberate and based on several methodological considerations. Firstly, the practice of partitioning samples into training and testing sets is a well-established approach in both machine learning and spatial analysis. This method ensures an unbiased assessment of model performance and generalizability using separate datasets (Chong et al., 2019; Oukawa et al., 2022).**

**Secondly, employing 100% samples for model training would preclude the availability of independent data for model validation, thereby increasing the risk of model overfitting. Moreover, a model trained with 100% samples would essentially be memorizing the training data rather than generalizing from it, thus compromising its predictive capabilities (Zhang and Yang, 2020).**

**Furthermore, models trained with the entire sampling dataset may not be robust, as they lack the rigor of validation against unseen data. In fact, the use of partitioned datasets for training and testing is widely adopted because it strikes an optimal balance between model accuracy and generalization (Verikas et al., 2011; Javeed et al., 2019; Yilmazer and Kocaman, 2020; Li et al., 2021; Meng et al., 2023; Zhan et al., 2023). Our analysis also corroborates this, demonstrating that a model trained with 70% of the samples has already achieved commendable performance, with an R-squared value of 0.955.**

**We hope you find these explanations rise to your expectations.**

**References**

Chong, D., Zhu, N., Luo, W., and Pan, X.: Human thermal risk prediction in indoor hyperthermal environments based on random forest, Sustainable Cities and Society, 49, 101595, 2019.

Javeed, A., Zhou, S., Yongjian, L., Qasim, I., Noor, A., and Nour, R.: An intelligent learning system based on random search algorithm and optimized random forest model for improved heart disease detection, IEEE access, 7, 180235-180243, 2019.

Li, X. H., Hou, J. L., and Huang, C. L.: High-Resolution Gridded Livestock Projection for Western China Based on Machine Learning, Remote. Sens., 13, 5038, https://doi.org/10.3390/rs13245038, 2021.

Liu, B. T.: Actual livestock carrying capacity estimation product in Qinghai-Tibet Plateau (2000-2019), National Tibetan Plateau Data Center. [Dataset], https://doi.org/10.11888/Ecolo.tpdc.271513, 2021.

Meng, N., Wang, L. J., Qi, W. C., Dai, X. H., Li, Z. Z., Yang, Y. Z., Li, R. N., Ma, J. F., and Zheng, H.: A high-resolution gridded grazing dataset of grassland ecosystem on the Qinghai-Tibet Plateau in 1982-2015, Sci. Data., 10, 68, https://doi.org/10.1038/s41597-023-01970-1, 2023.

Oukawa, G. Y., Krecl, P., and Targino, A. C.: Fine-scale modeling of the urban heat island: A comparison of multiple linear regression and random forest approaches, Sci. Total. Environ., 815, 152836, 2022.

Robinson, T. P., Wint, G. R., Conchedda, G., Van Boeckel, T. P., Ercoli, V., Palamara, E., Cinardi, G., D'Aietti, L., Hay, S. I., and Gilbert, M.: Mapping the global distribution of livestock, Plos. One., 9, e96084, https://doi.org/10.1371/journal.pone.0096084, 2014.

Verikas, A., Gelzinis, A., and Bacauskiene, M.: Mining data with random forests: A survey and results of new tests, Pattern recognition, 44, 330-349, 2011.

Zhan, N., Liu, W. H., Ye, T., Li, H. D., Chen, S., and Ma, H.: High-resolution livestock seasonal distribution data on the Qinghai-Tibet Plateau in 2020, Sci. Data., 10, 142, https://doi.org/10.1038/s41597-023-02050-0, 2023.

Zhang, F. and Yang, X.: Improving land cover classification in an urbanized coastal area by random forests: The role of variable selection, Remote. Sens. Environ., 251, 112105, 2020.

Yilmazer, S. and Kocaman, S.: A mass appraisal assessment study using machine learning based on multiple regression and random forest, Land. Use. Policy., 99, 104889, 2020.